# EFFI-LEARNER: Enhancing Efficiency of Generated Code via Self-Optimization

**Dong Huang***
The University of Hong Kong
dhuang@cs.hku.hk

**Jianbo Dai***
University of Edinburgh
j6dj6d@gmail.com

**Han Weng**
Beijing University of
Posts and Telecommunications
han.weng@bupt.edu.cn

**Puzhen Wu**
University College Dublin
puzhen.wu@ucdconnect.ie

**Yuhao Qing**
The University of Hong Kong
yhqing@cs.hku.hk

**Heming Cui**
The University of Hong Kong
Shanghai AI Laboratory
heming@cs.hku.hk

**Zhijiang Guo**[†]
University of Cambridge
zg283@cam.ac.uk

**Jie M. Zhang**
King's College London
jie.zhang@kcl.ac.uk

## Abstract

Large language models (LLMs) have shown remarkable progress in code generation, but their generated code often suffers from inefficiency, resulting in longer execution times and higher memory consumption. To address this issue, we propose **EFFI-LEARNER**, a self-optimization framework that utilizes execution overhead profiles to improve the efficiency of LLM-generated code. EFFI-LEARNER first generates code using an LLM, then executes it locally to capture execution time and memory usage profiles. These profiles are fed back to the LLM, which then revises the code to reduce overhead. To evaluate the effectiveness of EFFI-LEARNER, we conduct extensive experiments on the EffiBench, HumanEval, and MBPP with 16 open-source and 6 closed-source models. Our evaluation results demonstrate that through iterative self-optimization, EFFI-LEARNER significantly enhances the efficiency of LLM-generated code. For example, the execution time (ET) of StarCoder2-15B for the EffiBench decreases from 0.93 (s) to 0.12 (s) which reduces 87.1% execution time requirement compared with the initial code. The total memory usage (TMU) of StarCoder2-15B also decreases from 22.02 (Mb*s) to 2.03 (Mb*s), which decreases 90.8% total memory consumption during the execution process. The source code of EFFI-LEARNER was released in `https://github.com/huangd1999/EffiLearner`.

## 1 Introduction

Large language models (LLMs) have recently achieved significant advancements across various tasks [49, 4, 5, 42]. LLMs such as GPT-4 [50] and Copilot [44] have demonstrated considerable efficacy in code-related applications, including code completion [11, 6], debuggingg [25, 12], and translation [54, 1]. These innovative tools have been seamlessly integrated into popular development environments, significantly augmenting developer productivity by providing intelligent code

---

*Equal Contribution.
†Corresponding Author.

38th Conference on Neural Information Processing Systems (NeurIPS 2024).

**Task Description**

**Problem:** Given a non-empty array of integers nums, every element appears twice except for one. Find that single one. You must implement a solution with a linear runtime complexity and use only constant extra space.

**Example: Input:** nums = [4,1,2,1,2]    **Output:** 4

```
assert singleNumber([4, 1, 2, 1, 2]) == 4
```

**Initial Completion with Profile**

```
Timer unit: 1e-06 s
Total time: 0.0312331 s
Mem unit: MiB
Time   Mem usage    Line Contents
=============================================
             def singleNumber(nums):
  371.2 50.9     for i in range(len(nums)):
30860.9 50.9         if nums.count(nums[i]) == 1:
    0.9 50.9             return nums[i]
```

**EFFI-LEARNER Completion with Profile**

```
Timer unit: 1e-06 s
Total time: 0.000112618 s
Mem unit: MiB
Time   Mem usage    Line Contents
=============================================
             def singleNumber(nums):
  112.6 50.7     return reduce(operator.xor, nums)
```

Figure 1: A case for the task with code and EFFI-LEARNER refined version. The lower left panel shows the initial completion generated by an LLM, its profile shows its inefficiency. The lower right panel shows the final efficient answer output by applying EFFI-LEARNER.

recommendations based on natural language instructions. However, the primary focus of existing efforts has predominantly been on the correctness of the generated code [18], ensuring it meets the functional requirements and adheres to syntactical norms.

Despite advancements in ensuring code correctness, there remains a significant gap in the literature regarding the efficiency of code produced by LLMs. Efficiency is crucial as it translates to faster execution and lower memory and processing power consumption, which is especially important in resource-constrained environments such as mobile devices or embedded systems [18, 53, 13, 48, 17, 56, 9, 41, 52]. Recent studies [56, 48] reveal that LLM-generated code often exhibits lower efficiency in terms of execution time and memory usage when compared to canonical solutions. For instance, on a benchmark that evaluates efficiency, EffiBench [27], even the most powerful LLM (e.g., GPT-4-Turbo) generates code with suboptimal efficiency. The average and worst-case execution times are 1.69 and 45.49 times longer than those of the canonical solutions, respectively. This inefficiency underscores the need for developing new methods focused on evaluating and improving the efficiency of code generated by LLMs, ensuring that they produce correct and highly efficient code.

To bridge this gap, we draw inspiration from the methodology used by coders on coding platforms. When addressing a programming problem, coders typically write an initial program that is executable on the test cases. Next, they execute the code and obtain a profile of the execution time and memory usage overhead, as shown in Figure 1. Based on this overhead profile, the coder optimizes the code to enhance its efficiency. During this process, the coder extracts key information (e.g., execution times and memory usage of each line) from the overhead profile, which helps identify lines or operators that require significant overhead (e.g., loops that execute multiple times or unnecessary variables being saved). This information assists the coder in optimizing their code.

With this motivation, we propose **EFFI-LEARNER** to improve the efficiency of LLM-generated code. As shown in Figure 2, EFFI-LEARNER first requires the LLM to generate code based on the task description. Then, EFFI-LEARNER executes the generated code locally and captures the execution time and memory usage profile. These overhead profiles are fed back into the LLM, which then revises the code to reduce the overhead. Through multi-iteration self-optimization, the efficiency of the LLM-generated code is improved. While it's true that the iterative process requires extra time, it's crucial to recognize the long-term advantages that come with this investment. By optimizing the code, we can enhance the overall efficiency once it's deployed.

To evaluate the effectiveness of EFFI-LEARNER, we conduct extensive experiments on the EffiBench and two commonly used code generation benchmarks (i.e. HumanEval [11] and MBPP [6]) with 16 open-source and 6 closed-source models. We compare the efficiency of the code generated by the LLM before and after applying EFFI-LEARNER. The experimental results demonstrate that EFFI-LEARNER significantly improves the efficiency of the LLM-generated code. For example, the

execution time (ET) of StarCoder2-15B decreases from 0.93 (s) to 0.12 (s) which reduces 87.1% execution time requirement compared with the initial code. The max memory usage (MU) of DeepSeek-6.7B-Ins also decreased from 259.73 (Mb) to 36.97 (Mb), which reduces 85.8% max memory consumption for the code execution requirement. The total memory usage (TMU) of StarCoder2-15B also decreases from 22.02 (Mb*s) to 2.03 (Mb*s), which decreases 90.8% total memory consumption during the execution process.

## 2  Related Work

### 2.1  LLMs for Code Generation

The increasing popularity of LLMs for code generation has coincided with the growing availability of open-source code repositories and the need to boost developer productivity. Initial efforts focused on training models specifically for coding tasks, such as CodeT5 [61], AlphaCode [33], CodeGen [47], InCoder [19], StarCoder [32], SantaCoder [3] and DeepSeek Coder [15]. Contrastingly, models such as Codex [11] and CodeLLaMA [55] represent a subsequent stride, being finetuned from foundation models [8, 59]. These code LLMs have been applied to various tasks, including code generation [11, 14], program repair [25, 28], automated testing [31, 16], code translation [54, 1], type prediction [45, 64], and code summarization [26, 2]. Among these, code generation, where models generate code snippets based on natural language descriptions or docstrings, has become a critical domain for evaluating LLMs. While LLMs have achieved impressive results in code generation tasks like HumanEval [11] and MBPP [6], their efficiency has received less attention. Recent studies [56, 27, 48] have shown that LLM-generated code exhibits lower efficiency in execution time and memory usage compared to canonical solutions. These findings highlight the need for further research and development to improve the efficiency of LLM-generated code. In this work, we propose the first method that significantly improves the efficiency of code generated by a wide range of LLMs.

### 2.2  Learning From Feedback

A prevalent strategy for improving the behavior of LLMs is learning from feedback, mirroring human learning where individuals refine their actions through trial, error, and correction [7, 43]. Early efforts involve using human feedback to evaluate and refine models [30, 51, 21]. To minimize human intervention, another strategy focuses on automated feedback. These methods iteratively learn from automatically generated feedback signals, understanding the consequences of their actions and adapting their behaviors. The source of this automated feedback can be diverse, ranging from the LLM itself [40, 57], external tools [22, 37] or verifiers [36], external knowledge sources [20, 68] and even generation logits [67]. In code generation, the program executor is frequently used as a source of feedback for refining the model's initial code. For example, Self-Edit [69] and Self-Evolve [29] execute the initial program on example test cases and provide the execution results as feedback, prompting the LLM to refine the code. Self-Debug [12] explores using program explanation, unit tests, and program interpreters as feedback types. ALGO [70] employs a more fine-grained approach by generating a reference oracle program that solves the problem with an exhaustive search. Feedback is then collected by comparing the generated outputs with the oracle. While existing work primarily focuses on using feedback to edit the initial code to ensure correctness, our method explores using overhead profiles to improve the efficiency of the code.

## 3  EFFI-LEARNER

Inspired by the optimization strategies employed by human coders on coding platforms, we propose a framework EFFI-LEARNER to enhance the efficiency of LLM-generated code. Human coders typically analyze execution time and memory usage profiles to identify bottlenecks and optimize their code. EFFI-LEARNER leverages this principle by integrating a self-optimization loop into the code generation process. As illustrated in Figure 2, EFFI-LEARNER consists of three main components: **Code Generation**, **Overhead Profiling**, and **Code Refinement**, each playing a crucial role in the self-optimization process.

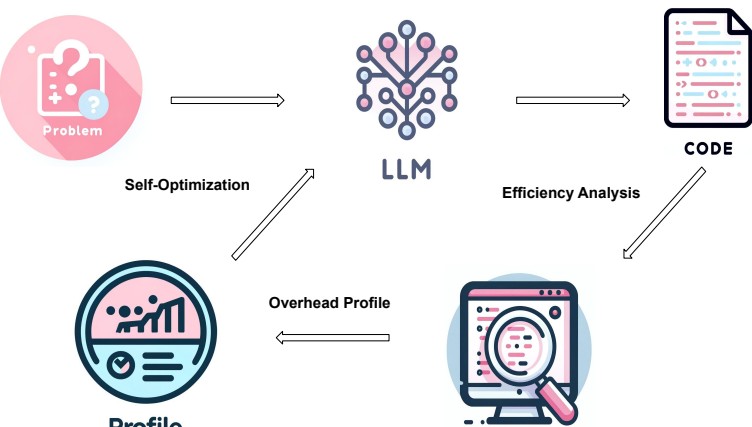

Figure 2: Pipeline of EFFI-LEARNER. LLMs first generate code for the given problem. This code is then executed locally to gather overhead profiles. These profiles are subsequently utilized by the LLMs to optimize the code in successive iterations, thereby enhancing the overall efficiency of the generated code. A comprehensive illustration is provided in the Appendix Figure 4-Figure 11.

## 3.1   Code Generation

Given a task description or code generation requirement, the LLM generates an initial version of the code. The LLM takes the task description as input and produces code that aims to solve the task.

## 3.2   Overhead Profiling

The generated code is executed locally to capture its execution time and memory usage overhead profiles. During this step, the code is run on a set of open test cases, and the execution time and memory usage for each line of code are recorded. This information forms the overhead profiles that provide insights into the efficiency of the generated code.

**Execution Time Profiling**  In this step, we measure the execution time of each line of code to identify potential bottlenecks and inefficiencies. To perform execution time profiling, we utilize the `line_profiler` library in Python. During the profiling process, we run the generated code on a set of open test cases provided by the dataset. The `line_profiler` library tracks the execution time of each line of code for all the test cases combined. This helps us assess the code's performance under different conditions and identify any performance bottlenecks. The execution time profiling results are reported based on the total consumption for all open test cases. The profiling output includes information such as the line number, the number of times each line is executed, and the total time spent on each line. These profiles serve as input for the subsequent code refinement step.

**Memory Usage Profiling**  Memory usage profiling is another essential aspect of the EFFI-LEARNER framework. It helps us understand how the generated code utilizes memory resources and identifies any memory-related inefficiencies or leaks. To profile memory usage, we employ the `memory_profiler` library in Python. During the memory usage profiling process, we run the generated code on the set of open test cases. The `memory_profiler` library monitors the memory usage of each line of code throughout the execution of all the test cases combined. It captures the memory usage at different points, such as before and after function calls, loop iterations, and memory allocation statements. The memory usage profiling results are reported based on the total consumption for all open test cases. The profiling output includes information such as the line number and the corresponding memory usage. These profiles provide valuable insights into the memory efficiency of the generated code and help identify areas for optimization.

## 3.3   Code Refinement

This component leverages the execution time and memory usage profiles to optimize the generated code. In this step, the LLM analyzes the overhead profiles to refine the code for better efficiency. To

enable self-optimization, we feed the overhead profiles back into the LLM along with the generated code. The LLM analyzes patterns in the overhead profiles, such as high execution time or excessive memory usage, and correlates them with specific code segments. It then applies optimization techniques, such as loop unrolling, memorization, data structure optimization, algorithm substitution, and code simplification, to improve the efficiency of the identified code segments.

During the self-optimization process, the LLM considers factors such as the impact of each optimization on the overall efficiency, the trade-offs between execution time and memory usage, and the preservation of code correctness. It aims to strike a balance between performance improvement and maintaining the functional integrity of the code. The LLM iteratively refines the code based on the overhead profiles, applying optimizations until a satisfactory level of efficiency is achieved or a predefined number of iterations is reached. The optimized code is then validated against the open test cases to ensure its functional correctness. By leveraging the execution time and memory usage profiles, the self-optimization step enables the LLM to improve the efficiency of the generated code.

### 3.4 Prompt Construction

We carefully design prompts to guide LLMs in optimizing code efficiency while ensuring the optimized code passes predefined test cases. The prompt template (Figure 3) used in EFFI-LEARNER's self-optimization stage includes a task description, test case, initial code, overhead analysis, and optimization rules. The task description provides context and requirements, the test case ensures correctness, and the initial code is the starting point for optimization. The overhead analysis highlights performance metrics and areas for improvement, while the optimization rules focus the LLM on enhancing efficiency, encapsulating the optimized code, and excluding the test case from the code block. This comprehensive prompt equips the LLM with the necessary information to effectively optimize code, maintain consistency across models and tasks, and facilitate comparison of their code optimization capabilities, advancing the field of LLM-driven code optimization. Details of the template can be found in Appendex A.3.

## 4 Evaluation

### 4.1 Dataset and Metrics

We evaluate EFFI-LEARNER on EffiBench [27]. Following their settings, we use Execution Time (ET), Normalized Execution Time (NET), Max Memory Usage (MU), Normalized Max Memory Usage (NMU), Total Memory Usage (TMU), and Normalized Total Memory Usage (NTMU) as metrics. We provide a detailed definition of these metrics in A.5. Following the setup of EffiBench, evaluation metrics were only calculated on the tasks that generated code for both the initial version and EFFI-LEARNER optimized code that can pass all private test cases provided by the dataset[3].

Following Huang et al. [27], we utilize the open test cases to calculate the efficiency metrics during the self-optimization process, while private test cases provided by EffiBench were used for the final result evaluation. For HumanEval and MBPP datasets, we set the test cases provided by HumanEval and MBPP as open test cases, while test cases provided by EvalPlus [35] (i.e., HumanEval-Plus, MBPP-Plus) as private test cases that were used to calculate the final results.

### 4.2 Implementation Details

All of the experiments are conducted in an edge server with an Intel Xeon Platinum 8336C CPU with 128 cores, and 8 * NVIDIA A100-SXM GPUs Total memory capacity of 2.0TiB.

**Models** We evaluate EFFI-LEARNER's effectiveness on both open-source and closed-source LLMs[4]. For open-source models, we evaluate EFFI-LEARNER with OpenCodeInterpreter (1.3B, 6.7B, and 33B) [72], DeepSeek-Instruct (1.3B, 6.7B, and 33B) [24], CodeLlama (7B, 13B, 34B, and 70B) [55], XwinCoder (7B and 34B) [58], StarCoder (3B, 7B, and 15B) [32], and WizardCoder-13B [38], where the detailed versions of LLMs are demonstrated in supplementary file. For closed-source models, we

---

[3]Some LLMs may not generate correct initial code, we do not report the efficiency results for it.
[4]We provide results for more LLMs in Table 5 in Appendix.

Table 1: Code efficiency of LLMs with EFFI-LEARNER on EffiBench. The percentage in the brackets indicates the extent of the reduction for each respective item. Top performing LLMs are highlighted.

| Model | ET (s) | NET | MU (Mb) | NMU | TMU (Mb*s) | NTMU |
|---|---|---|---|---|---|---|
| Open-source LLMs | | | | | | |
| OpenCodeInterpreter-1.3B | 1.60 | 1.52 | 38.91 | 1.00 | 89.16 | 1.11 |
| | 1.29 (19.4%) | 1.23 (19.1%) | 38.91 (0.0%) | 1.00 (0.0%) | 70.63 (20.8%) | 0.88 (20.7%) |
| OpenCodeInterpreter-6.7B | 0.34 | 2.41 | 36.82 | 1.00 | 13.36 | 1.56 |
| | 0.28 (17.6%) | 1.91 (20.7%) | 38.60 (-4.8%) | 1.00 (0.0%) | 14.16 (-6.0%) | 1.44 (7.7%) |
| OpenCodeInterpreter-33B | 0.29 | 2.10 | 35.48 | 1.00 | 13.06 | 1.93 |
| | 0.28 (3.4%) | 2.00 (4.8%) | 36.30 (-2.3%) | 1.00 (0.0%) | 11.54 (11.6%) | 1.64 (15.0%) |
| DeepSeek-1.3B-Ins | 1.42 | 1.32 | 36.04 | 1.00 | 40.61 | 1.12 |
| | 1.15 (19.0%) | 1.07 (18.9%) | 36.04 (0.0%) | 1.00 (0.0%) | 35.48 (12.6%) | 0.98 (12.5%) |
| DeepSeek-6.7B-Ins | 0.37 | 2.60 | 259.73 | 7.25 | 555.18 | 67.70 |
| | 0.34 (8.1%) | 2.37 (8.8%) | 36.97 (**85.8%**) | 1.00 (**86.2%**) | 13.66 (97.5%) | 1.46 (97.8%) |
| DeepSeek-33B-Ins | 0.29 | 2.21 | 34.53 | 1.06 | 14.44 | 2.91 |
| | 0.25 (13.8%) | 1.84 (16.7%) | 32.67 (5.4%) | 0.99 (6.6%) | 8.15 (43.6%) | 1.55 (46.7%) |
| CodeLlama-7B | 4.70 | 3.68 | 46.76 | 0.99 | 212.41 | 1.93 |
| | 4.52 (3.8%) | 3.54 (3.8%) | 38.67 (17.3%) | 0.82 (17.2%) | 157.76 (25.7%) | 1.43 (25.9%) |
| CodeLlama-13B | 2.45 | 2.19 | 42.46 | 0.93 | 137.40 | 1.51 |
| | 2.28 (6.9%) | 2.04 (6.8%) | 42.12 (0.8%) | 0.93 (0.0%) | 119.36 (13.1%) | 1.31 (13.2%) |
| CodeLlama-34b | 1.05 | 7.75 | 57.57 | 1.70 | 94.79 | 15.65 |
| | 1.02 (2.9%) | 7.34 (5.3%) | 40.62 (29.4%) | 1.11 (34.7%) | 52.12 (45.0%) | 7.02 (55.1%) |
| CodeLlama-70b | 0.52 | 3.93 | 109.61 | 3.57 | 203.92 | 54.15 |
| | 0.47 (9.6%) | 3.84 (2.3%) | 26.42 (75.9%) | 1.00 (72.0%) | 14.53 (92.9%) | 6.52 (88.0%) |
| XwinCoder-7B | 2.80 | 2.81 | 55.54 | 1.52 | 208.23 | 3.47 |
| | 2.43 (13.2%) | 2.44 (13.2%) | 49.10 (11.6%) | 1.34 (11.8%) | 158.20 (24.0%) | 2.64 (23.9%) |
| XwinCoder-34B | 0.77 | 5.68 | 49.77 | 1.49 | 61.36 | 12.11 |
| | 0.69 (10.4%) | 5.11 (10.0%) | 52.12 (-4.7%) | 1.47 (1.3%) | 57.89 (5.7%) | 9.92 (18.1%) |
| StarCoder2-3B | 1.10 | 1.25 | 24.31 | 1.00 | 17.47 | 1.19 |
| | 1.02 (7.3%) | 1.15 (8.0%) | 24.28 (0.1%) | 1.00 (0.0%) | 16.38 (6.2%) | 1.12 (5.9%) |
| StarCoder2-7B | 3.69 | 5.34 | 26.42 | 1.08 | 82.38 | 7.62 |
| | 2.99 (19.0%) | 4.32 (19.1%) | 26.40 (0.1%) | 1.08 (0.0%) | 68.61 (16.7%) | 6.35 (16.7%) |
| StarCoder2-15B | 0.93 | 7.58 | 26.35 | 1.00 | 22.02 | 10.88 |
| | 0.12 (**87.1%**) | 1.03 (**86.4%**) | 27.67 (-5.0%) | 1.01 (-1.0%) | 2.03 (**90.8%**) | 1.06 (**90.3%**) |
| WizardCoder-13B | 3.43 | 2.11 | 86.72 | 1.35 | 324.83 | 1.92 |
| | 2.93 (14.6%) | 1.80 (14.7%) | 71.02 (18.1%) | 1.11 (17.8%) | 219.69 (32.4%) | 1.30 (32.3%) |
| Closed-source LLMs | | | | | | |
| GPT-3.5-Turbo-0301 | 0.36 | 2.50 | 91.25 | 2.45 | 157.50 | 19.75 |
| | 0.28 (**22.2%**) | 2.01 (**19.6%**) | 36.08 (**60.5%**) | 0.99 (**59.6%**) | 12.43 (**92.1%**) | 1.64 (**91.7%**) |
| GPT-3.5-Turbo-1106 | 0.28 | 1.96 | 36.12 | 1.01 | 12.79 | 1.73 |
| | 0.26 (7.1%) | 1.90 (3.1%) | 34.02 (5.8%) | 1.00 (1.0%) | 11.41 (10.8%) | 1.62 (6.4%) |
| GPT-4-Turbo-Preview | 0.27 | 1.96 | 33.94 | 1.00 | 11.82 | 1.89 |
| | 0.25 (7.4%) | 1.88 (4.1%) | 33.17 (2.3%) | 1.00 (0.0%) | 10.18 (13.9%) | 1.76 (6.9%) |
| GPT-4 | 0.31 | 2.19 | 80.88 | 2.26 | 129.91 | 17.90 |
| | 0.28 (9.7%) | 2.06 (5.9%) | 63.82 (21.1%) | 1.83 (19.0%) | 80.74 (37.8%) | 11.86 (33.7%) |
| Claude-3-Haiku | 0.36 | 2.51 | 48.33 | 1.30 | 52.67 | 6.73 |
| | 0.33 (8.3%) | 2.30 (8.4%) | 37.37 (22.7%) | 1.03 (20.8%) | 17.18 (67.4%) | 2.37 (64.8%) |
| Claude-3-Sonnet | 0.42 | 2.90 | 60.46 | 1.62 | 82.52 | 10.12 |
| | 0.35 (16.7%) | 2.47 (14.8%) | 42.31 (30.0%) | 1.17 (27.8%) | 28.95 (64.9%) | 3.76 (62.8%) |

use GPT-3.5-Turbo, GPT-4-Turbo, GPT-4 [50], Claude-3-haiku, and Claude-3-sonnet[5]. These LLMs have achieved competitive pass@1 scores in various code generation tasks [11, 6, 35].

**Setup** We first collect the generated code from each LLM and evaluate its correctness using open test cases. Only the code that passes all test cases is considered for efficiency evaluation. This approach ensures consistency in the evaluated tasks across different self-optimization iterations, as EFFI-LEARNER focuses on improving the efficiency of initially correct code without altering its pass@1 score. By evaluating a diverse set of open-source and closed-source LLMs, we aim to provide a comprehensive assessment of the efficiency of LLM-generated code and the effectiveness of EFFI-LEARNER in improving code efficiency across different models and architectures.

## 4.3 Main Results

**Open-source LLMs** As shown in Table 1, we observe that the efficiency metrics for all models have been increased in most experiments once we apply EFFI-LEARNER to optimize the efficiency of LLM-generated code. For example, in OpenCodeInterpreter-1.3B, the execution time for its generated code decreases from 1.60 (s) to 1.29 (s), a reduction of 19.4% in execution time. Additionally, the

---

[5]We do not include Claude-3-opus in our experiments due to limited resources.

Table 2: Effect of the number of self-optimization steps in EFFI-LEARNER.

| Steps | ET (s) | NET | MU (Mb) | NMU | TMU (Mb*s) | NTMU |
|---|---|---|---|---|---|---|
| CodeLlama-70B | | | | | | |
| 0 | 0.52 | 3.93 | 109.61 | 3.57 | 203.92 | 54.15 |
| 1 | 0.48 (7.7%) | 3.94 (-0.3%) | 26.47 (75.9%) | 1.00 (72.0%) | 14.91 (92.7%) | 6.69 (87.6%) |
| 2 | 0.48 (7.7%) | 3.89 (1.0%) | 26.47 (75.9%) | 1.00 (72.0%) | 14.69 (92.8%) | 6.60 (87.8%) |
| 3 | 0.47 (9.6%) | 3.85 (2.0%) | 26.42 (75.9%) | 1.00 (72.0%) | 14.60 (92.8%) | 6.56 (87.9%) |
| 4 | 0.47 (9.6%) | 3.84 (2.3%) | 26.42 (75.9%) | 1.00 (72.0%) | 14.54 (92.9%) | 6.53 (87.9%) |
| 5 | 0.47 (9.6%) | 3.84 (2.3%) | 26.42 (75.9%) | 1.00 (72.0%) | 14.53 (92.9%) | 6.52 (88.0%) |
| GPT-3.5-Turbo-0301 | | | | | | |
| 0 | 0.36 | 2.50 | 91.25 | 2.45 | 157.50 | 19.75 |
| 1 | 0.33 (8.3%) | 2.35 (6.0%) | 36.09 (60.4%) | 0.99 (59.6%) | 13.70 (91.3%) | 1.81 (90.8%) |
| 2 | 0.31 (13.9%) | 2.18 (12.8%) | 36.09 (60.4%) | 0.99 (59.6%) | 13.04 (91.7%) | 1.72 (91.3%) |
| 3 | 0.29 (19.4%) | 2.06 (17.6%) | 36.08 (60.5%) | 0.99 (59.6%) | 12.57 (92.0%) | 1.66 (91.6%) |
| 4 | 0.29 (19.4%) | 2.03 (18.8%) | 36.08 (60.5%) | 0.99 (59.6%) | 12.50 (92.1%) | 1.65 (91.6%) |
| 5 | 0.28 (22.2%) | 2.01 (19.6%) | 36.08 (60.5%) | 0.99 (59.6%) | 12.43 (92.1%) | 1.64 (91.7%) |

TMU of OpenCodeInterpreter-1.3B decreases from 89.16 (Mb*s) to 70.63 (Mb*s). Furthermore, in certain edge cases, EFFI-LEARNER significantly enhances efficiency. For example, the ET of StarCoder2-15B decreases from 0.93 (s) to 0.12 (s) and the NET also decreases from 7.48 to 1.03, reducing execution time requirements by 87.1% compared to the initial code. The MU and NMU of DeepSeek-6.7B-Ins also decrease from 259.73 (Mb) and 7.25 to 36.97 (Mb) and 1.06, reducing the maximum memory consumption by 85.8% for the code execution requirement. Moreover, we can also observe that the TMU and NTMU of StarCoder2-15B also decrease from 22.02 (Mb*s) and 10.88 to 2.03 (Mb*s) and 1.06, which decreases 90.8% memory consumption during the execution process. These results demonstrate the effectiveness of EFFI-LEARNER in optimizing the code generated by open-source LLMs.

**Closed-source LLMs** Similar to open-source LLMs, we observe that the efficiency metrics for most closed-source LLMs have been improved after applying EFFI-LEARNER to optimize the efficiency of the generated code. For instance, the execution time for code generated by GPT-3.5-Turbo-0301 decreases from 0.36 (s) to 0.28 (s), reducing the execution time by 22.2%. The MU and NMU of GPT-3.5-Turbo-0301 also decrease from 91.25 (Mb) and 2.45 to 36.08 (Mb) and 0.99, respectively, which reduces the max memory consumption for code execution by 60.5%. Furthermore, the TMU and NTMU of GPT-3.5-Turbo-0301 decrease from 157.50 (Mb*s) and 19.75 to 12.43 (Mb*s) and 1.64, respectively, decreasing memory consumption during the execution process by 92.1%.

The improvements in efficiency metrics across both open-source and closed-source LLMs highlight the generalizability and adaptability of EFFI-LEARNER. By iteratively refining the generated code based on efficiency profiler feedback, EFFI-LEARNER enables LLMs to produce more efficient code without compromising the correctness of the generated solutions. The consistent improvements across various models and architectures demonstrate the potential of EFFI-LEARNER as a model-agnostic approach for optimizing the efficiency of LLM-generated code in real-world applications.

## 4.4 Impact of Self-Optimization Steps

To investigate the impact of the number of self-optimization steps on the efficiency of the EFFI-LEARNER-optimized code, we conduct an ablation study by varying the number of steps from 0 to 5. Table 2 for CodeLlama-70B and GPT-3.5-Turbo-0301 at different self-optimization steps.

**CodeLlama-70B** We can observe that after the first self-optimization step, the MU decreases from 109.61 (Mb) to 26.47 (Mb), reducing 75.9% maximum memory requirement compared with the initial code generated by CodeLlama-70B. Similarly, the TMU decreases from 54.15 (Mb*s) to 6.69 (Mb*s), reducing 87.6% of memory consumption during code execution. As the number of steps increases, the efficiency metrics gradually improve. By the fifth step, the ET reaches 0.47 (s), reducing the 1.9% execution time requirement compared with the first-step generated code, and the TMU settles at 14.53, reducing 0.2% total memory usage from the first step.

**GPT-3.5-Turbo-0301** Similar to CodeLlama-70B, the MU decreases from 91.25 (Mb) to 36.09 (Mb) after the first self-optimization step, reducing 60.4% maximum memory requirement compared with the initial code. The TMU also shows a substantial reduction from 157.50 (Mb*s) to 13.70 (Mb*s),

Table 3: Contribution of different components in EFFI-LEARNER. We evaluate how different feedback profilers affect the efficiency of LLM-generated code. Unsupervised self-refine only requires LLMs to optimize the efficiency of the code. Result-Aware Self-Refine feedback the ET, MU, and TMU to the LLMs and require it to improve the efficiency. Memory Profiler and Execution Time Profiler feedback the memory profiler and execution time profiler to the LLMs and then LLMs can based on the profile optimize the efficiency of the code.

| Optimization Profile | ET (s) | NET | MU (Mb) | NMU | TMU (Mb*s) | NTMU |
|---|---|---|---|---|---|---|
| CodeLlama-70B | | | | | | |
| Initial Version | 0.52 | 3.93 | 109.61 | 3.57 | 203.92 | 54.15 |
| Unsupervised Self-Refine | 0.79 (-51.9%) | 6.87 (-74.8%) | 279.41 (-154.9%) | 10.58 (-196.4%) | 1261.83 (-518.8%) | 600.95 (-1009.8%) |
| Result-Aware Self-Refine | 0.79 (-51.9%) | 6.87 (-74.8%) | 282.57 (-157.8%) | 10.70 (-199.7%) | 1270.93 (-523.2%) | 605.29 (-1017.8%) |
| Memory Profiler | 0.53 (-1.9%) | 4.34 (-10.4%) | 26.38 (75.9%) | 0.99 (72.3%) | 15.77 (92.3%) | 7.06 (87.0%) |
| Execution Time Profiler | 0.51 (1.9%) | 4.17 (-6.1%) | 26.44 (75.9%) | 1.00 (72.0%) | 15.53 (92.4%) | 6.97 (87.1%) |
| EFFI-LEARNER | 0.47 (9.6%) | 3.84 (2.3%) | 26.42 (75.9%) | 1.00 (72.0%) | 14.53 (92.9%) | 6.52 (88.0%) |
| GPT-3.5-Turbo-0301 | | | | | | |
| Initial Version | 0.36 | 2.50 | 91.25 | 2.45 | 157.50 | 19.75 |
| Unsupervised Self-Refine | 0.32 (11.1%) | 2.46 (1.6%) | 78.39 (14.1%) | 2.12 (13.5%) | 312.99 (-98.7%) | 42.42 (-114.8%) |
| Result-Aware Self-Refine | 0.30 (16.7%) | 2.25 (10.0%) | 58.65 (35.7%) | 1.61 (34.3%) | 195.49 (-24.1%) | 27.16 (-37.5%) |
| Memory Profiler | 0.34 (5.6%) | 2.40 (4.0%) | 36.85 (59.6%) | 1.00 (59.2%) | 16.34 (89.6%) | 2.10 (89.4%) |
| Execution Time Profiler | 0.33 (8.3%) | 2.34 (6.4%) | 36.43 (60.1%) | 0.99 (59.6%) | 14.07 (91.1%) | 1.81 (90.8%) |
| EFFI-LEARNER | 0.28 (22.2%) | 2.01 (19.6%) | 36.08 (60.5%) | 0.99 (59.6%) | 12.43 (92.1%) | 1.64 (91.7%) |

reducing 91.3% of memory consumption during code execution. As the number of steps increases, the efficiency metrics continue to improve steadily. By the fifth step, the ET reaches 0.28 (s), reducing the 15.2% execution time requirement compared with the first-step generated code, and the TMU settles at 12.43 (Mb*s), reducing 9.3% total memory usage from the first step.

Table 2 demonstrates the significant impact of the number of self-optimization steps on the efficiency of the EFFI-LEARNER-optimized code. For both CodeLlama-70B and GPT-3.5-Turbo-0301, the first self-optimization step yields the most substantial improvements in code efficiency. As the number of steps increases, the efficiency metrics continue to improve, albeit with diminishing returns. By the fifth step, the efficiency metrics reach their lowest values, demonstrating the effectiveness of EFFI-LEARNER's iterative self-optimization approach in enhancing the efficiency of LLM-generated code. The evaluation results highlight that the majority of efficiency improvements occur in the first few steps, with subsequent steps contributing to further refinements of the optimized code.

## 4.5 Feedback of Overhead Profile

To show the effectiveness of the overhead profile in guiding LLMs to refine their generated code, we compare the performance of EFFI-LEARNER with two alternative approaches: Unsupervised Self-Refine and Result-Aware Self-Refine [40, 57]. Unsupervised Self-Refine uses a prompt that directly requires the LLM to refine the code without providing additional information. Result-Aware Self-Refine feeds the ET, MU, and TMU, then requires the LLM to refine the code based on these metrics. Table 3 presents the code efficiency metrics for CodeLlama-70B and GPT-3.5-Turbo-0301 using different code refinement approaches.

**CodeLlama-70B** Unsupervised Self-Refine and Result-Aware Self-Refine result in significant increases in ET, memory usage (MU), and TMU compared to the initial version. Unsupervised Self-Refine increases ET by 51.9%, MU by 154.9%, and TMU by 518.8%, while Result-Aware Self-Refine increases ET by 51.9%, MU by 157.8%, and TMU by 523.2%. In contrast, EFFI-LEARNER incorporates the overhead profile feedback and achieves a 9.6% reduction in ET, a 75.9% reduction in MU, and a 92.9% reduction in TMU compared to the initial version.

**GPT-3.5-Turbo-0301** Unsupervised Self-Refine and Result-Aware Self-Refine show some improvements in ET and MU compared to the initial version. Unsupervised Self-Refine reduces ET by 11.1% and MU by 14.1%, while Result-Aware Self-Refine reduces ET by 16.7% and MU by 35.7%. However, both approaches lead to substantial increases in TMU, with Unsupervised Self-Refine increasing TMU by 98.7% and Result-Aware Self-Refine increasing TMU by 24.1%. On the other hand, EFFI-LEARNER achieves a 22.2% reduction in ET, a 60.5% reduction in MU, and a 92.1% reduction in TMU compared to the initial version.

These results highlight the importance of the overhead profile feedback in guiding LLMs to generate more efficient code. Without the overhead profile, the code refinement process using alternative

Table 4: Results of EFFI-LEARNER on HumanEval dataset, where we evaluate CodeLlama family generated code's efficiency. Full results are listed in Appendix Table 8.

| Steps | ET (s) | NET | MU (Mb) | NMU | TMU (Mb*s) | NTMU |
|---|---|---|---|---|---|---|
| CodeLlama-7b | 0.20 | 0.71 | 57.39 | 0.91 | 7.08 | 0.70 |
| | 0.18 (10.0%) | 0.63 (11.3%) | 57.07 (0.6%) | 0.91 (0.0%) | 6.18 (12.7%) | 0.61 (12.9%) |
| CodeLlama-13b | 0.23 | 0.95 | 58.13 | 0.96 | 7.97 | 0.94 |
| | 0.20 (13.0%) | 0.80 (15.8%) | 58.03 (0.2%) | 0.96 (0.0%) | 6.64 (16.7%) | 0.79 (16.0%) |
| CodeLlama-34b | 0.24 | 0.95 | 61.79 | 1.01 | 8.45 | 0.96 |
| | 0.21 (12.5%) | 0.81 (14.7%) | 61.55 (0.4%) | 1.00 (1.0%) | 6.99 (17.3%) | 0.80 (16.7%) |
| CodeLlama-70b | 0.21 | 0.93 | 60.19 | 1.01 | 6.76 | 1.01 |
| | 0.18 (14.3%) | 0.79 (15.1%) | 59.49 (1.2%) | 1.00 (1.0%) | 5.75 (14.9%) | 0.86 (14.9%) |

prompts fails to improve code efficiency and even leads to significant performance degradation. The overhead profile provides valuable insights into the resource consumption of the generated code, enabling LLMs to make targeted optimizations and achieve substantial efficiency improvements.

## 4.6 Discussion

Table 5: Code efficiency of widely-studied LLMs reported by EFFI-LEARNER.

| Model | ET (s) | NET | MU (Mb) | NMU | TMU (Mb*s) | NTMU |
|---|---|---|---|---|---|---|
| Phind-CodeLlama-34B-v2 | 0.52 | 3.28 | 157.16 | 3.36 | 337.30 | 24.44 |
| | 0.40 (23.1%) | 2.51 (23.5%) | 68.27 (56.6%) | 1.45 (56.8%) | 65.64 (80.5%) | 4.86 (80.1%) |
| Artigenz-Coder-DS-6.7B | 0.39 | 2.75 | 65.73 | 1.70 | 95.65 | 10.87 |
| | 0.32 (17.9%) | 2.30 (16.4%) | 59.00 (10.2%) | 1.62 (4.7%) | 79.67 (16.7%) | 10.73 (1.3%) |
| Magicoder-S-DS-6.7B | 0.22 | 1.59 | 40.19 | 1.09 | 17.58 | 2.28 |
| | 0.21 (4.5%) | 1.50 (5.7%) | 38.29 (4.7%) | 1.07 (1.8%) | 15.27 (13.1%) | 2.22 (2.6%) |
| Mistral-7B-codealpaca-lora | 2.36 | 18.40 | 28.88 | 1.00 | 57.92 | 24.36 |
| | 1.45 (38.6%) | 12.17 (33.9%) | 27.45 (5.0%) | 1.03 (-3.0%) | 35.46 (38.8%) | 17.28 (29.1%) |
| CodeFuse-DeepSeek-33B | 0.40 | 3.10 | 70.39 | 2.06 | 191.15 | 32.20 |
| | 0.39 (2.5%) | 3.01 (2.9%) | 63.22 (10.2%) | 1.85 (10.2%) | 156.81 (18.0%) | 26.42 (18.0%) |
| CodeLlama-34b-hf | 2.08 | 15.68 | 46.41 | 1.26 | 128.46 | 17.87 |
| | 1.95 (6.3%) | 14.67 (6.4%) | 46.40 (0.0%) | 1.26 (0.0%) | 125.22 (2.5%) | 17.42 (2.5%) |
| speechless-starcoder2-15b | 0.19 | 1.74 | 27.39 | 0.99 | 3.20 | 1.75 |
| | 0.13 (31.6%) | 1.19 (31.6%) | 27.25 (0.5%) | 0.99 (0.0%) | 2.17 (32.2%) | 1.19 (32.0%) |
| gpt-3.5-turbo-0613 | 0.56 | 4.32 | 35.48 | 1.00 | 20.11 | 3.00 |
| | 0.49 (12.5%) | 3.75 (13.2%) | 35.47 (0.0%) | 1.00 (0.0%) | 17.84 (11.3%) | 2.66 (11.3%) |

**Generalizability across benchmarks** In Table 1, we evaluated EFFI-LEARNER's effectiveness on the EffiBench dataset. To illustrate EFFI-LEARNER's generalizability in other datasets, we conduct experiments on the HumanEval and MBPP datasets in Appendix Table 8 and Table 9. We also provide EFFI-LEARNER's effectiveness on the HumanEval dataset in CodeLlama models in Table 4. We can

Table 6: Pass@1 of LLMs generated initial code and EFFI-LEARNER optimized code.

| Model | Initial Pass@1 | EFFI-LEARNER Pass@1 |
|---|---|---|
| OpenCodeInterpreter-DS-1.3B | 5.8 | 5.4 |
| OpenCodeInterpreter-DS-6.7B | 13.6 | 13.2 |
| OpenCodeInterpreter-DS-33B | 24.7 | 24.4 |
| deepseek-1.3b-Ins | 4.8 | 4.5 |
| deepseek-6.7b-Ins | 7.2 | 7.0 |
| deepseek-33b-Ins | 10.0 | 9.9 |
| CodeLlama-7b | 7.0 | 7.0 |
| CodeLlama-13b | 9.7 | 9.6 |
| CodeLlama-34b | 13.5 | 13.0 |
| CodeLlama-70b | 7.8 | 7.4 |
| XwinCoder-13B | 10.5 | 10.2 |
| XwinCoder-34B | 21.2 | 21.2 |
| starcoder2-3b | 1.6 | 1.2 |
| starcoder2-7b | 1.9 | 1.8 |
| starcoder2-15b | 0.7 | 0.4 |
| WizardCoder-13B | 4.0 | 3.9 |

observe that the coding efficiency of CodeLlama and other LLMs (See Table 8) also increases when we utilize EFFI-LEARNER to optimize LLM-generated code. For example, the ET of CodeLlama-70B decreases from 0.21 (s) to 0.18 (s), which reduces 14.3% execution time. As shown in Table 8 and Table 9, results demonstrate that EFFI-LEARNER can consistently improve the efficiency of LLM-generated code for other datasets.

**Generalizability across LLMs** In Table 1, we evaluate EFFI-LEARNER's effectiveness on six types of open-source LLMs. To illustrate EFFI-LEARNER's generalizability in other LLMs, we also conduct experiments on other LLMs in Table 5. Our evaluation results demonstrate that EFFI-LEARNER can improve the efficiency of LLM-generated code for different LLMs. For example, the execution time of Mistral-7B-codealpaca-lora decreases from 2.36 (s) to 1.45 (s), which reduces 38.6% execution time compared with the initial code. The total memory usage of Phind-CodeLlama-34B-v2 also decreases from 337.30 (Mb*s) to 65.64 (Mb*s), which reduces 80.5% total memory requirement.

**Impact on correctness** We provide the pass@1 of LLM-generated initial code and EFFI-LEARNER optimized code for EffiBench in Table 6. We observe that the pass@1 of EFFI-LEARNER optimized code may be lower than LLM-generated initial code. The key reason is that during the self-optimization process, EFFI-LEARNER only uses public test cases to guide code efficiency optimization for correct initial code. However, since public test cases may not cover all edge cases in the private test cases (test cases used to evaluate pass@1 of LLMs), this can cause the pass@1 of EFFI-LEARNER generated code to be lower than the initial code. Nevertheless, we observe that the pass@1 of EFFI-LEARNER only decreases by about 0% to 0.5%, which means that only a few of the codes will be incorrect. As shown in Table 1, we can observe that the code efficiency is largely increased. We believe that this minor decrease in pass@1 is worthwhile considering the significant efficiency gains.

**Case study** To illustrate how EFFI-LEARNER improves the efficiency of LLM-generated code, we provide a case illustration in Appendix Figure 4-Figure 11. As shown in Figure 6, we can observe that the execution time of the initial code is 23.59 (s) while in the self-optimized code, the execution time decreases from 23.59 (s) to 3.36 (s). The key reason is that in the initial code, the algorithm uses a standard unidirectional Breadth-First Search (BFS), which explores all possible states level by level starting from the initial state. This method results in a large number of states to explore, leading to significant computational overhead. In contrast, the self-optimized code employs a bidirectional BFS, which simultaneously searches from both the initial state and the target state. This reduces the search space by meeting in the middle, significantly decreasing the number of states that need to be explored and thereby improving the execution time.

**Error Analysis** We also provide a case illustration to explain why some code efficiency does not improve significantly when EFFI-LEARNER is applied to LLM-generated code. As shown in Appendix Figure 12-Figure 18, we observe that the initial code only requires 0.0012 (s) to execute, while in the optimized code, the execution time is still 0.0011 (s). The key reason for this minimal improvement is that both implementations already operate with the same theoretical time complexity of $O(\log(\min(m, n)))$. Given the problem's constraints and small input sizes, the actual runtime differences are overshadowed by the inherent efficiency of the binary search algorithm. Additionally, the overhead of function calls and Python runtime operations can further minimize the observed performance gains. Therefore, while the optimized code may offer clearer partition management and slight improvements, the overall efficiency remains largely unchanged due to the already optimized nature of the initial approach.

# 5    Conclusion

This paper focuses on the critical issue of efficiency in code generated by LLMs. While LLMs have shown impressive capabilities in code generation, their output often suffers from suboptimal efficiency, leading to slower execution and higher resource consumption. To tackle this challenge, we propose EFFI-LEARNER, a novel self-optimization framework that leverages execution overhead profiles to guide LLMs in improving code efficiency. Extensive experiments and analysis demonstrate that EFFI-LEARNER significantly enhances the efficiency of LLM-generated code, achieving substantial reductions in execution time and memory usage. For future work, we would like to investigate the application of EFFI-LEARNER to other programming tasks and languages, as well as explore the potential benefits of incorporating domain-specific knowledge into the optimization process.

# 6 ACKNOWLEDGMENT

The work is supported in part by National Key R&D Program of China (2022ZD0160201), HK RGC RIF (R7030-22), HK ITF (GHP/169/20SZ), a Huawei Flagship Research Grant in 2023, HK RGC GRF (Ref: 17208223 & 17204424), and the HKU-CAS Joint Laboratory for Intelligent System Software.

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

# A Appendix

## A.1 Limitations

EFFI-LEARNER presents a compelling solution for enhancing the efficiency of code generated by LLMs. However, it's crucial to acknowledge several potential limitations. Firstly, the multi-iteration self-optimization process at the heart of EFFI-LEARNER can be time-consuming, particularly when applied to intricate programming tasks. Yet, it's important to note that this investment of time can yield significant long-term benefits, as the optimized codes, once deployed, can considerably improve efficiency. Moreover, the process of feeding overhead profiles to LLMs to prompt code optimization may consume more tokens. This is due to the fact that the length of the overhead profiles necessitates additional tokens. Lastly, the effectiveness of EffiLearner has been primarily evaluated on Python. Therefore, its performance in different programming languages or environments may vary, underscoring the need for further testing and validation of this approach in a diverse range of contexts.

## A.2 Broader Impacts

**Positive Societal Impacts** The proposed method EFFI-LEARNER has the potential to bring about several positive societal impacts. Primarily, it can significantly enhance productivity by enabling developers to complete tasks more quickly and efficiently, due to the faster execution times and lower memory and processing power consumption of the optimized code. This is particularly beneficial in resource-constrained environments, such as mobile devices or embedded systems, where conserving resources is crucial. Moreover, EFFI-LEARNER's focus on improving code efficiency can lead to more cost-effective solutions, as lower resource consumption translates to reduced operational costs.

**Negative Societal Impacts** The increasing effectiveness of LLM-based tools like EFFI-LEARNER could also have negative societal impacts. For instance, it could lead to an over-reliance on these systems, potentially resulting in a decline in human coding skills and a lack of understanding of the underlying code. This could be problematic in situations where human intervention is necessary. Additionally, there is a risk of job displacement in the coding and programming industry, as the demand for human coders may decrease, particularly for routine coding tasks. Lastly, while EFFI-LEARNER can improve code efficiency, it does not necessarily address potential security vulnerabilities or privacy issues in the code, which could have significant societal impacts if not properly managed.

## A.3 Prompt Template

```
prompt = f"""
Please optimize the efficiency of the following Python code based on the task description, test case,
↪  and overhead analysis provided. Ensure the optimized code can pass the given test case.

Task Description:
{task_description}

Test Case:
{test_case}

Original Code:
```python
{completion}
```

Overhead Analysis:
{overhead_prompt}
```

Figure 3: Prompt template used by EFFI-LEARNER in the self-optimization stage.

Table 7: Evaluation results of EFFI-LEARNER and baselines. Since the finetuned link for GPT-3.5-turbo from PIE is not available, we use the fine-tuned CodeLlama 7B for experiments. Due to the fine-tuned PIE CodeLlama 7B does not have the same correct tasks as the original CodeLlama, we then do not provide the initial version for the experiments.

| OptimizationProfile | ET(s) | NET | MU(Mb) | NMU | TMU(Mb*s) | NTMU |
|---|---|---|---|---|---|---|
| GPT-3.5-Turbo-0301 | | | | | | |
| InitialVersion | 0.36 | 2.50 | 91.25 | 2.45 | 157.50 | 19.75 |
| UnsupervisedSelf-Refine | 0.32 | 2.46 | 78.39 | 2.12 | 312.99 | 42.42 |
| Result-AwareSelf-Refine | 0.30 | 2.25 | 58.65 | 1.61 | 195.49 | 27.16 |
| Self-Edit | 0.42 | 3.67 | 59.86 | 1.65 | 24.87 | 3.28 |
| Critic | 0.60 | 4.25 | 102.75 | 2.82 | 351.91 | 46.39 |
| DirectlyEfficiency | 0.43 | 3.03 | 59.11 | 1.67 | 20.37 | 2.88 |
| Self-RefineEfficiency | 0.40 | 2.83 | 59.11 | 1.67 | 18.80 | 2.65 |
| IsSelf-Refine | 0.40 | 2.88 | 61.83 | 1.81 | 36.29 | 5.69 |
| Self-Reasoning | 0.89 | 6.21 | 60.64 | 1.62 | 45.91 | 5.61 |
| Self-Relfection | 0.81 | 5.67 | 60.64 | 1.62 | 39.35 | 4.80 |
| EFFI-LEARNER | 0.28 (22.2%) | 2.01 (19.6%) | 36.08 (60.5%) | 0.99 (59.6%) | 12.43 (92.1%) | 1.64 (91.7%) |
| CodeLlama7B(PIE:HQ+SelfPlay) | | | | | | |
| PIE+Zero-Shot | 0.87 | 5.73 | 74.83 | 1.81 | 109.29 | 9.69 |
| PIE+EFFI-LEARNER +Zero-Shot | 0.79 | 5.41 | 65.78 | 1.68 | 89.90 | 7.84 |
| PIE+Few-Shot | 0.82 | 5.58 | 73.57 | 1.74 | 98.02 | 8.92 |
| PIE+EFFI-LEARNER +Few-Shot | 0.41 | 2.97 | 73.10 | 1.74 | 59.69 | 5.09 |
| PIE+CoT | 0.79 | 5.14 | 73.14 | 1.74 | 63.93 | 5.35 |
| PIE+EFFI-LEARNER +COT | 0.45 | 2.84 | 71.15 | 1.71 | 58.06 | 4.77 |
| PIE+DynamicRetrieval,k=4 | 0.74 | 5.36 | 68.64 | 1.51 | 85.24 | 7.78 |
| PIE+EFFI-LEARNER +DynamicRetrieval,k=4 | 0.41 | 3.36 | 68.63 | 1.51 | 52.34 | 4.52 |
| Supersonic | | | | | | |
| Supersonic | 1.40 | 10.33 | 113.06 | 3.18 | 329.59 | 56.24 |
| Supersonic+EFFI-LEARNER | 1.34 | 9.91 | 102.26 | 2.87 | 267.47 | 45.64 |

## A.4 Comparison with Baselines

To demonstrate EFFI-LEARNER's effectiveness with existing baselines and some prompt engineering methods that can refine the prompt from the correctness to the efficiency of LLM-generated code.

Table 8: Evaluation results of EFFI-LEARNER's effectiveness in the HumanEval dataset.

| Steps | ET (s) | NET | MU (Mb) | NMU | TMU (Mb*s) | NTMU |
|---|---|---|---|---|---|---|
| OpenCodeInterpreter-DS-1.3B | 0.20 | 0.86 | 57.24 | 1.00 | 6.63 | 0.84 |
| | 0.19 (5.0%) | 0.81 (5.8%) | 57.17 (0.1%) | 1.00 (0.0%) | 6.20 (6.5%) | 0.79 (6.0%) |
| OpenCodeInterpreter-DS-6.7B | 0.21 | 0.98 | 58.83 | 1.06 | 6.79 | 0.99 |
| | 0.21 (0.0%) | 0.97 (1.0%) | 58.79 (0.1%) | 1.06 (0.0%) | 6.64 (2.2%) | 0.97 (2.0%) |
| OpenCodeInterpreter-DS-33B | 0.21 | 0.95 | 59.90 | 1.05 | 7.05 | 0.94 |
| | 0.21 (0.0%) | 0.93 (2.1%) | 59.93 (-0.1%) | 1.05 (0.0%) | 6.87 (2.6%) | 0.92 (2.1%) |
| deepseek-coder-1.3b-instruct | 0.23 | 0.90 | 62.80 | 1.00 | 7.85 | 0.87 |
| | 0.22 (4.3%) | 0.85 (5.6%) | 62.96 (-0.3%) | 1.00 (0.0%) | 7.80 (0.6%) | 0.86 (1.1%) |
| deepseek-coder-6.7b-instruct | 0.22 | 0.76 | 59.57 | 1.00 | 7.34 | 0.77 |
| | 0.19 (13.6%) | 0.68 (10.5%) | 59.72 (-0.3%) | 1.00 (0.0%) | 6.59 (10.2%) | 0.69 (10.4%) |
| deepseek-coder-33b-instruct | 0.21 | 0.95 | 63.52 | 0.99 | 7.18 | 0.95 |
| | 0.20 (4.8%) | 0.92 (3.2%) | 63.49 (0.0%) | 0.99 (0.0%) | 6.99 (2.6%) | 0.92 (3.2%) |
| CodeLlama-7b-Instruct-hf | 0.20 | 0.71 | 57.39 | 0.91 | 7.08 | 0.70 |
| | 0.18 (10.0%) | 0.63 (11.3%) | 57.07 (0.6%) | 0.91 (0.0%) | 6.18 (12.7%) | 0.61 (12.9%) |
| CodeLlama-13b-Instruct-hf | 0.23 | 0.95 | 58.13 | 0.96 | 7.97 | 0.94 |
| | 0.20 (13.0%) | 0.80 (15.8%) | 58.03 (0.2%) | 0.96 (0.0%) | 6.64 (16.7%) | 0.79 (16.0%) |
| CodeLlama-34b-Instruct-hf | 0.24 | 0.95 | 61.79 | 1.01 | 8.45 | 0.96 |
| | 0.21 (12.5%) | 0.81 (14.7%) | 61.55 (0.4%) | 1.00 (1.0%) | 6.99 (17.3%) | 0.80 (16.7%) |
| CodeLlama-70b-Instruct-hf | 0.21 | 0.93 | 60.19 | 1.01 | 6.76 | 1.01 |
| | 0.18 (14.3%) | 0.79 (15.1%) | 59.49 (1.2%) | 1.00 (1.0%) | 5.75 (14.9%) | 0.86 (14.9%) |
| XwinCoder-13B | 0.27 | 1.08 | 61.14 | 1.04 | 9.25 | 1.09 |
| | 0.25 (7.4%) | 1.01 (6.5%) | 61.15 (-0.0%) | 1.04 (0.0%) | 8.62 (6.8%) | 1.02 (6.4%) |
| XwinCoder-34B | 0.25 | 1.07 | 60.75 | 1.05 | 8.46 | 1.08 |
| | 0.22 (12.0%) | 0.93 (13.1%) | 60.75 (0.0%) | 1.05 (0.0%) | 7.33 (13.4%) | 0.94 (13.0%) |
| WizardCoder-7B | 0.21 | 0.91 | 58.59 | 1.01 | 6.63 | 0.89 |
| | 0.18 (14.3%) | 0.79 (13.2%) | 57.97 (1.1%) | 1.00 (1.0%) | 5.79 (12.7%) | 0.78 (12.4%) |
| WizardCoder-13B | 0.21 | 0.81 | 60.59 | 1.00 | 7.22 | 0.79 |
| | 0.19 (9.5%) | 0.73 (9.9%) | 60.53 (0.1%) | 1.00 (0.0%) | 6.47 (10.4%) | 0.71 (10.1%) |
| WizardCoder-34B | 0.22 | 0.79 | 58.13 | 1.00 | 7.10 | 0.78 |
| | 0.17 (22.7%) | 0.62 (21.5%) | 58.42 (-0.5%) | 1.00 (0.0%) | 5.46 (23.1%) | 0.60 (23.1%) |
| starcoder2-3b | 0.24 | 1.02 | 62.45 | 1.00 | 7.73 | 0.89 |
| | 0.19 (20.8%) | 0.79 (22.5%) | 62.69 (-0.4%) | 1.00 (0.0%) | 6.68 (13.6%) | 0.77 (13.5%) |
| starcoder2-7b | 0.21 | 0.89 | 62.53 | 1.00 | 7.41 | 0.85 |
| | 0.19 (9.5%) | 0.78 (12.4%) | 62.85 (-0.5%) | 1.00 (0.0%) | 6.40 (13.6%) | 0.74 (12.9%) |

Table 9: Evaluation results of EFFI-LEARNER's effectiveness in the MBPP dataset

| Steps | ET (s) | NET | MU (Mb) | NMU | TMU (Mb*s) | NTMU |
|---|---|---|---|---|---|---|
| OpenCodeInterpreter-DS-1.3B | 0.28 | 0.94 | 59.01 | 1.01 | 11.73 | 0.98 |
| | 0.25 (10.7%) | 0.84 (10.6%) | 58.99 (0.0%) | 1.01 (0.0%) | 10.59 (9.7%) | 0.89 (9.2%) |
| OpenCodeInterpreter-DS-6.7B | 0.26 | 1.06 | 58.39 | 1.00 | 9.25 | 1.08 |
| | 0.21 (19.2%) | 0.87 (17.9%) | 58.37 (0.0%) | 1.00 (0.0%) | 7.10 (23.2%) | 0.83 (23.1%) |
| OpenCodeInterpreter-DS-33B | 0.44 | 1.59 | 58.72 | 1.00 | 20.19 | 1.86 |
| | 0.31 (29.5%) | 1.14 (28.3%) | 58.70 (0.0%) | 1.00 (0.0%) | 13.22 (34.5%) | 1.22 (34.4%) |
| deepseek-coder-1.3b-instruct | 0.63 | 1.68 | 354.01 | 6.05 | 1463.46 | 89.12 |
| | 0.62 (1.6%) | 1.64 (2.4%) | 339.91 (4.0%) | 5.81 (4.0%) | 1414.13 (3.4%) | 86.11 (3.4%) |
| deepseek-coder-6.7b-instruct | 0.76 | 3.62 | 58.44 | 1.00 | 39.11 | 5.69 |
| | 0.21 (72.4%) | 0.98 (72.9%) | 58.34 (0.2%) | 1.00 (0.0%) | 6.67 (82.9%) | 0.97 (83.0%) |
| deepseek-coder-33b-instruct | 0.58 | 2.33 | 53.48 | 0.91 | 28.74 | 3.16 |
| | 0.19 (67.2%) | 0.75 (67.8%) | 53.34 (0.3%) | 0.91 (0.0%) | 5.88 (79.5%) | 0.65 (79.4%) |
| CodeLlama-7b-Instruct-hf | 0.45 | 2.04 | 56.96 | 0.97 | 13.26 | 1.79 |
| | 0.42 (6.7%) | 1.89 (7.4%) | 56.78 (0.3%) | 0.97 (0.0%) | 11.98 (9.7%) | 1.62 (9.5%) |
| CodeLlama-13b-Instruct-hf | 0.53 | 2.11 | 55.37 | 0.95 | 21.75 | 2.34 |
| | 0.52 (1.9%) | 2.04 (3.3%) | 55.29 (0.1%) | 0.95 (0.0%) | 21.13 (2.9%) | 2.28 (2.6%) |
| CodeLlama-34b-Instruct-hf | 0.42 | 1.18 | 69.80 | 1.19 | 84.01 | 5.47 |
| | 0.41 (2.4%) | 1.13 (4.2%) | 69.32 (0.7%) | 1.19 (0.0%) | 74.78 (11.0%) | 4.87 (11.0%) |
| CodeLlama-70b-Instruct-hf | 0.23 | 1.06 | 58.13 | 0.98 | 7.65 | 1.05 |
| | 0.20 (13.0%) | 0.93 (12.3%) | 58.05 (0.1%) | 0.98 (0.0%) | 6.67 (12.8%) | 0.91 (13.3%) |
| XwinCoder-7B | 0.23 | 1.14 | 58.45 | 1.00 | 7.19 | 1.10 |
| | 0.18 (21.7%) | 0.90 (21.1%) | 58.44 (0.0%) | 1.00 (0.0%) | 5.89 (18.1%) | 0.90 (18.2%) |
| XwinCoder-13B | 0.50 | 1.96 | 58.38 | 1.00 | 23.88 | 2.50 |
| | 0.41 (18.0%) | 1.61 (17.9%) | 58.34 (0.1%) | 1.00 (0.0%) | 18.95 (20.6%) | 1.98 (20.8%) |
| XwinCoder-34B | 0.38 | 1.44 | 58.27 | 1.00 | 14.77 | 1.48 |
| | 0.35 (7.9%) | 1.32 (8.3%) | 58.22 (0.1%) | 1.00 (0.0%) | 13.54 (8.3%) | 1.36 (8.1%) |
| WizardCoder-Python-7B-V1.0-GPTQ | 0.22 | 1.05 | 58.44 | 0.99 | 7.19 | 1.03 |
| | 0.20 (9.1%) | 0.93 (11.4%) | 58.33 (0.2%) | 0.99 (0.0%) | 6.41 (10.8%) | 0.91 (11.7%) |
| WizardCoder-Python-13B-V1.0-GPTQ | 0.62 | 1.35 | 57.74 | 0.99 | 30.66 | 1.43 |
| | 0.59 (4.8%) | 1.28 (5.2%) | 57.70 (0.1%) | 0.99 (0.0%) | 29.56 (3.6%) | 1.38 (3.5%) |
| WizardCoder-Python-34B-V1.0-GPTQ | 0.68 | 2.43 | 56.75 | 0.97 | 34.06 | 3.14 |
| | 0.65 (4.4%) | 2.33 (4.1%) | 56.78 (-0.1%) | 0.97 (0.0%) | 32.63 (4.2%) | 3.01 (4.1%) |
| starcoder2-3b | 0.17 | 0.83 | 45.82 | 0.79 | 5.10 | 0.77 |
| | 0.16 (5.9%) | 0.80 (3.6%) | 43.46 (5.2%) | 0.74 (6.3%) | 4.69 (8.0%) | 0.70 (9.1%) |
| starcoder2-7b | 1.72 | 8.63 | 25.61 | 0.44 | 40.42 | 6.22 |
| | 1.72 (0.0%) | 8.61 (0.2%) | 25.56 (0.2%) | 0.44 (0.0%) | 40.19 (0.6%) | 6.19 (0.5%) |
| starcoder2-15b | 0.19 | 1.05 | 58.62 | 1.01 | 6.23 | 1.05 |
| | 0.18 (5.3%) | 0.99 (5.7%) | 58.14 (0.8%) | 1.00 (1.0%) | 5.92 (5.0%) | 1.00 (4.8%) |

## A.5 Detailed Evaluation Metric for Efficiency

In our work, we adopt the efficiency metrics proposed by EffiBench [27] to evaluate the effectiveness of EFFI-LEARNER in improving the efficiency of LLM-generated code.

**Execution Time (ET)**    Execution time (ET) measures the average time taken for code execution. Mathematically, ET is defined as:

$$ET = \frac{1}{N} \sum^{N} T_{\text{code}}$$

where $ET$ is the execution time metric, $T_{\text{code}}$ is the execution time of the code (with all the test cases), and $N$ is the number of codes generated by code generation models used for evaluation.

**Normalized Execution Time (NET)**    Normalized Execution Time (NET) measures the execution time required by generated code relative to that of a canonical solution. We define NET as:

$$NET = \frac{1}{N} \sum^{N} \frac{T_{\text{code}}}{T_{\text{canonical}}}$$

where $T_{\text{code}}$ is the execution time of the generated code, and $T_{\text{canonical}}$ is the execution time of the canonical solution. A NET value greater than 1 indicates that the generated code is slower than the canonical solution, while a value less than 1 suggests the generated code is faster.

**Max Memory Usage (MU)**    Max Memory Usage (MU) measures the average max memory consumption during code execution. Mathematically, MU is defined as:

$$MU = \frac{1}{N} \sum^{N} M_{\text{code}}$$

where $MU$ is the memory usage metric, $M_{\text{code}}$ is the max memory consumption of the generated code among all the test cases, and $N$ is the number of code instances generated by code generation models used for evaluation. This metric is critical for assessing the resource efficiency of generated code, particularly in environments with limited maximum memory capacity.

**Normalized Max Memory Usage (NMU)**    Normalized Max Memory Usage (NMU) quantifies how the max memory efficiency of the generated code compares to the canonical solution. We define NMU as:

$$NMU = \frac{1}{N} \sum^{N} \frac{M_{\text{code}}}{M_{\text{canonical}}}$$

where $NMU$ is the normalized max memory usage metric, $M_{\text{code}}$ is the max memory usage of the generated code, and $M_{\text{canonical}}$ is the max memory usage of the canonical solution. An NMU value less than 1 indicates that the generated code is more memory-efficient than the canonical solution, whereas a value greater than 1 suggests it is less efficient in terms of memory usage. This metric provides a relative measure of the memory optimization in the generated code in comparison to a standard baseline.

**Total Memory Usage (TMU)**    Total Memory Usage (TMU) assesses the efficiency of memory usage throughout the execution of code, taking into account both the magnitude and duration of memory utilization. To calculate TMU, first, monitor and record the memory usage at discrete time intervals during the execution, resulting in a memory usage profile $M(t)$, where $t$ represents time. Then, compute the area under the curve of $M(t)$ over the total execution time, $T_{\text{total}}$, using numerical integration methods such as the trapezoidal rule:

$$TMU = \frac{1}{N} \sum^{N} \int_{0}^{T_{\text{total}}} M(t) \, dt$$

A lower TMU value indicates higher memory efficiency, reflecting an optimized balance between the amount of memory used and the duration of its usage.

**Normalized Total Memory Usage (NTMU)**    The Normalized Total Memory Usage (NTMU) offers a comparison of the dynamic memory efficiency between the generated code and the canonical solution. To determine NTMU, calculate the TMU for both the generated code and the canonical solution. Normalize the TMU of the generated code by dividing it by the TMU of the canonical solution:

$$NTMU = \frac{1}{N} \sum^{N} \frac{TMU_{\text{code}}}{TMU_{\text{canonical}}}$$

where $TMU_{\text{code}}$ is the TMU of the generated code and $TMU_{\text{canonical}}$ is the TMU of the canonical solution. An NTMU value less than 1 signifies that the generated code manages dynamic memory more efficiently compared to the canonical solution, while a value greater than 1 indicates a less efficient management of dynamic memory. This metric provides insight into the relative use of dynamic memory of generated code compared to an established benchmark.

## A.6   Additional Related Work

**Instruction Tuning for Code**    Instruction tuning has proven effective in enhancing the usability and overall performance of LLMs across various language tasks [51, 63, 60, 71]. This approach has been extended to the domain of code generation. The core challenge is the acquisition of high-quality instructional data, which is often labor-intensive. To address this, recent research has focused on developing methods to generate synthetic instruction data. Studies have shown that textbook-quality synthetic data alone can improve a model's coding and reasoning capabilities [23, 34]. One early effort was Self-Instruct [62], which utilized LLMs to generate synthetic instruction-response pairs using carefully crafted prompts. The same LLM was then instruction-tuned on this synthetic data. Code Alpaca [10] applied the Self-Instruct approach with GPT models, tailoring it specifically for code generation, editing, and optimization tasks. Building upon this, WizardCoder [39] adapted the Evol-Instruct technique [66] to the coding domain by designing heuristic prompts to create more complex and diverse synthetic data. OSS-Instruct [65] took a different approach by leveraging LLMs to automatically generate new coding problems inspired by random code snippets from open-source repositories. In contrast, Octopack [46] focused on collecting and filtering high-quality Git commit messages that resemble natural language instructions.

**Problem:** You have a lock in front of you with 4 circular wheels. Each wheel has 10 slots: '0', '1', '2', '3', '4', '5', '6', '7', '8', '9'. The wheels can rotate freely and wrap around: for example we can turn '9' to be '0', or '0' to be '9'. Each move consists of turning one wheel one slot.
The lock initially starts at '0000', a string representing the state of the 4 wheels.
You are given a list of deadends dead ends, meaning if the lock displays any of these codes, the wheels of the lock will stop turning and you will be unable to open it.
Given a target representing the value of the wheels that will unlock the lock, return the minimum total number of turns required to open the lock, or -1 if it is impossible.

**Example 1:**
 **Input:** deadends = ["0201","0101","0102","1212","2002"], target = "0202"
 **Output:** 6
 **Explanation:** A sequence of valid moves would be "0000" $->$ "1000" $->$ "1100" $->$ "1200" $->$ "1201" $->$ "1202" $->$ "0202". Note that a sequence like "0000" $->$ "0001" $->$ "0002" $->$ "0102" $->$ "0202" would be invalid, because the wheels of the lock become stuck after the display becomes the dead end "0102".

**Example 2:**
 **Input:** deadends = ["8888"], target = "0009"
 **Output:** 1
 **Explanation:** We can turn the last wheel in reverse to move from "0000" $->$ "0009".

**Example 3:**
 **Input:** deadends = ["8887","8889","8878","8898","8788","8988","7888","9888"], target = "8888"
 **Output:** -1
 **Explanation:** We cannot reach the target without getting stuck.

**Constraints:**

- $1 \leq$ deadends.length $\leq 500$
- deadends[i].length == 4
- target.length == 4
- target will not be in the list deadends.
- target and deadends[i] consist of digits only.

```
solution = Solution()
assert solution.openLock(["0201","0101","0102","1212","2002"], "0202") == 6
assert solution.openLock(["8888"], "0009") == 1
assert solution.openLock(["8887","8889","8878","8898","8788","8988","7888","9888"], "8888") == -1
```

Figure 4: A case illustration of GPT-4-turbo-preview (OpenLock). Part A: Task Description.

```python
from typing import List
from collections import deque

class Solution:
    def openLock(self, deadends: List[str], target: str) -> int:
        def neighbors(node):
            for i in range(4):
                x = int(node[i])
                for d in (-1, 1):
                    y = (x + d) % 10
                    yield node[:i] + str(y) + node[i+1:]

        dead = set(deadends)
        queue = deque([('0000', 0)])
        seen = {'0000'}

        while queue:
            node, depth = queue.popleft()
            if node == target:
                return depth
            if node in dead:
                continue
            for nei in neighbors(node):
                if nei not in seen:
                    seen.add(nei)
                    queue.append((nei, depth + 1))

        return -1
```

Figure 5: A case illustration of GPT-4-turbo-preview (OpenLock). Part B: GPT-4-turbo-preview generated code.

```
Execution Time Profile

Total time: 23.5891 s
File: 752_tmp.py
Function: openLock at line 24

Line #      Hits         Time  Per Hit   % Time  Line Contents
==============================================================
    24                                               @profile
    25                                               def openLock(self, deadends: List[str], target:
    ↪  str) -> int:
    26       200      14151.9     70.8      0.1          @profile
    27       200        107.6      0.5      0.0          def neighbors(node):
    28                                                       for i in range(4):
    29                                                           x = int(node[i])
    30                                                           for d in (-1, 1):
    31                                                               y = (x + d) % 10
    32                                                               yield node[:i] + str(y) +
    ↪  node[i+1:]
    33
    34       100        189.1      1.9      0.0          dead = set(deadends)
    35       100        125.8      1.3      0.0          queue = deque([('0000', 0)])
    36       100         58.8      0.6      0.0          seen = {'0000'}
    37
    38    478172     109953.5      0.2      0.5          while queue:
    39    478172     183034.8      0.4      0.8              node, depth = queue.popleft()
    40    478172     113871.5      0.2      0.5              if node == target:
    41       100         18.8      0.2      0.0                  return depth
    42    478072     105351.0      0.2      0.4              if node in dead:
    43       253         44.3      0.2      0.0                  continue
    44   4300371   21651001.1      5.0     91.8              for nei in neighbors(node):
    45   3822552     969789.3      0.3      4.1                  if nei not in seen:
    46    576544     200262.9      0.3      0.8                      seen.add(nei)
    47    576544     241153.3      0.4      1.0                      queue.append((nei, depth + 1))
    48
    49                                                   return -1

Total time: 0.0129757 s
File: 752_tmp.py
Function: neighbors at line 26

Line #      Hits         Time  Per Hit   % Time  Line Contents
==============================================================
    26                                               @profile
    27                                               def neighbors(node):
    28      5345       1206.2      0.2      9.3          for i in range(4):
    29      4276       1595.9      0.4     12.3              x = int(node[i])
    30     12828       2305.9      0.2     17.8              for d in (-1, 1):
    31      8552       2248.4      0.3     17.3                  y = (x + d) % 10
    32      8552       5619.3      0.7     43.3                  yield node[:i] + str(y) +
    ↪  node[i+1:]
```

Figure 6: A case illustration of GPT-4-turbo-preview (OpenLock). Part C: Execution time profile.

**Memory Usage Profile**

```
Line #    Mem usage    Increment   Occurrences   Line Contents
=============================================================
    14    48.5 MiB    48.5 MiB            1      @profile(stream=profile_stream,
    ↪  precision=PROFILE_PRECISION)
    15                                           def openLock(self, deadends: List[str], target:
    ↪  str) -> int:
    16    48.8 MiB     0.0 MiB         4225          def neighbors(node):
    17    48.8 MiB     0.0 MiB        21123              for i in range(4):
    18    48.8 MiB     0.0 MiB        16898                  x = int(node[i])
    19    48.8 MiB     0.0 MiB        50696                  for d in (-1, 1):
    20    48.8 MiB     0.0 MiB        33797                      y = (x + d) % 10
    21    48.8 MiB     0.1 MiB        67594                      yield node[:i] + str(y) +
    ↪  node[i+1:]
    22
    23    48.5 MiB     0.0 MiB            1          dead = set(deadends)
    24    48.5 MiB     0.0 MiB            1          queue = deque([('0000', 0)])
    25    48.5 MiB     0.0 MiB            1          seen = {'0000'}
    26
    27    48.8 MiB     0.0 MiB         4229          while queue:
    28    48.8 MiB     0.1 MiB         4229              node, depth = queue.popleft()
    29    48.8 MiB     0.0 MiB         4229              if node == target:
    30    48.8 MiB     0.0 MiB            1                  return depth
    31    48.8 MiB     0.0 MiB         4228              if node in dead:
    32                                                       continue
    33    48.8 MiB     0.0 MiB        38022              for nei in neighbors(node):
    34    48.8 MiB     0.0 MiB        33797                  if nei not in seen:
    35    48.8 MiB     0.2 MiB         4977                      seen.add(nei)
    36    48.8 MiB     0.0 MiB         4977                      queue.append((nei, depth + 1))
    37
    38                                               return -1
```

Figure 7: A case illustration of GPT-4-turbo-preview (OpenLock). Part D: Memory usage profile.

**EFFI-LEARNER Completion**

```python
from typing import List
from collections import deque

class Solution:
    def openLock(self, deadends: List[str], target: str) -> int:
        """Calculate the minimum number of moves to open the lock from '0000' to the target avoiding
        ↪   deadends."""
        if '0000' in deadends:
            return -1
        if target == '0000':
            return 0

        def getNextStates(state: str) -> List[str]:
            """Generate the next states for a given lock state."""
            res = []
            for i in range(4):
                current_digit = int(state[i])
                for diff in [-1, 1]:  # decrease or increase the digit
                    new_digit = (current_digit + diff) % 10
                    res.append(state[:i] + str(new_digit) + state[i+1:])
            return res

        def bfs(queue: Deque[str], distances: dict, other_distances: dict) -> int:
            """Perform a step of BFS, expanding the current queue."""
            for _ in range(len(queue)):
                current = queue.popleft()
                step_count = distances[current]
                for nextState in getNextStates(current):
                    if nextState in deadends_set or nextState in distances:
                        continue
                    if nextState in other_distances:
                        return step_count + 1 + other_distances[nextState]
                    distances[nextState] = step_count + 1
                    queue.append(nextState)
            return -1

        deadends_set = set(deadends)
        # Early exit if the initial state is a deadend
        if '0000' in deadends_set:
            return -1

        # Initialize BFS structures for bidirectional search
        queue1, queue2 = deque(['0000']), deque([target])
        distance1, distance2 = {"0000": 0}, {target: 0}

        # Perform BFS from both ends
        while queue1 and queue2:
            result = bfs(queue1, distance1, distance2)
            if result != -1:
                return result
            result = bfs(queue2, distance2, distance1)
            if result != -1:
                return result

        return -1
```

Figure 8: A case illustration of GPT-4-turbo-preview (OpenLock). Part E: Self-Optimization Code.

```
Execution Time Profile

Total time: 3.35559 s
File: 752_tmp.py
Function: openLock at line 24

Line #      Hits         Time  Per Hit   % Time  Line Contents
==============================================================
    24                                           @profile
    25                                           def openLock(self, deadends: List[str], target:
   ↪  str) -> int:
    26                                               """Calculate the minimum number of moves to
   ↪  open the lock from '0000' to the target avoiding deadends."""
    27       100         65.1      0.7      0.0       if '0000' in deadends:
    28                                                   return -1
    29       100         35.0      0.3      0.0       if target == '0000':
    30                                                   return 0
    31

    57
    58       100         99.3      1.0      0.0       deadends_set = set(deadends)
    59                                               # Early exit if the initial state is a
   ↪  deadend
    60       100         36.7      0.4      0.0       if '0000' in deadends_set:
    61                                                   return -1
    62
    63                                               # Initialize BFS structures for bidirectional
   ↪  search
    64       100        116.8      1.2      0.0       queue1, queue2 = deque(['0000']),
   ↪  deque([target])
    65       100         47.6      0.5      0.0       distance1, distance2 = {"0000": 0}, {target:
   ↪  0}
    66
    67                                               # Perform BFS from both ends
    68       518        154.2      0.3      0.0       while queue1 and queue2:
    69       518    1936061.9   3737.6     57.7           result = bfs(queue1, distance1,
   ↪  distance2)
    70       518        153.0      0.3      0.0           if result != -1:
    71        52         10.3      0.2      0.0               return result
    72       466    1388682.7   2980.0     41.4           result = bfs(queue2, distance2,
   ↪  distance1)
    73       466        136.6      0.3      0.0           if result != -1:
    74        48          9.5      0.2      0.0               return result
    75
    76                                               return -1
```

Figure 9: A case illustration of GPT-4-turbo-preview (OpenLock). Part F1: Execution time profile for Self-Optimization Code.

## Execution Time Profile

```
Total time: 0.000697948 s
File: 752_tmp.py
Function: getNextStates at line 32

Line #      Hits         Time  Per Hit   % Time  Line Contents
==============================================================
    32                                               @profile
    33                                               def getNextStates(state: str) -> List[str]:
    34                                                   """Generate the next states for a given
↪   lock state."""
    35        52         12.1      0.2      1.7           res = []
    36       260         62.4      0.2      8.9           for i in range(4):
    37       208         74.4      0.4     10.7               current_digit = int(state[i])
    38       624        127.4      0.2     18.2               for diff in [-1, 1]:  # decrease or
↪   increase the digit
    39       416        104.5      0.3     15.0                   new_digit = (current_digit +
↪   diff) % 10
    40       416        308.4      0.7     44.2                   res.append(state[:i] +
↪   str(new_digit) + state[i+1:])
    41        52          8.8      0.2      1.3           return res

Total time: 0.00182689 s
File: 752_tmp.py
Function: bfs at line 43

Line #      Hits         Time  Per Hit   % Time  Line Contents
==============================================================
    43                                               @profile
    44                                               def bfs(queue: Deque[str], distances: dict,
↪   other_distances: dict) -> int:
    45                                                   """Perform a step of BFS, expanding the
↪   current queue."""
    46        57         12.9      0.2      0.7           for _ in range(len(queue)):
    47        52         14.0      0.3      0.8               current = queue.popleft()
    48        52         11.4      0.2      0.6               step_count = distances[current]
    49       462       1499.4      3.2     82.1               for nextState in
↪   getNextStates(current):
    50       411        114.9      0.3      6.3                   if nextState in deadends_set or
↪   nextState in distances:
    51       235         34.5      0.1      1.9                       continue
    52       176         38.7      0.2      2.1                   if nextState in other_distances:
    53         1          0.7      0.7      0.0                       return step_count + 1 +
↪   other_distances[nextState]
    54       175         47.5      0.3      2.6                   distances[nextState] = step_count
↪   + 1
    55       175         51.9      0.3      2.8                   queue.append(nextState)
    56         5          0.9      0.2      0.0           return -1
```

Figure 10: A case illustration of GPT-4-turbo-preview (OpenLock). Part F2: Execution time profile for Self-Optimization Code.

```
Line #    Mem usage    Increment  Occurrences   Line Contents
=================================================================
    14     48.4 MiB     48.4 MiB          1       @profile(stream=profile_stream,
↪   precision=PROFILE_PRECISION)
    15                                             def openLock(self, deadends: List[str], target:
↪   str) -> int:
    16                                                 """Calculate the minimum number of moves to
↪   open the lock from '0000' to the target avoiding deadends."""
    17     48.4 MiB      0.0 MiB          1           if '0000' in deadends:
    18                                                     return -1
    19     48.4 MiB      0.0 MiB          1           if target == '0000':
    20                                                     return 0
    21
    22     48.4 MiB      0.0 MiB        819           def getNextStates(state: str) -> List[str]:
    23                                                     """Generate the next states for a given
↪   lock state."""
    24     48.4 MiB      0.0 MiB        818               res = []
    25     48.4 MiB      0.0 MiB       4090               for i in range(4):
    26     48.4 MiB      0.0 MiB       3272                   current_digit = int(state[i])
    27     48.4 MiB      0.0 MiB       9816                   for diff in [-1, 1]:  # decrease or
↪   increase the digit
    28     48.4 MiB      0.0 MiB       6544                       new_digit = (current_digit + diff)
↪   % 10
    29     48.4 MiB      0.0 MiB       6544                       res.append(state[:i] +
↪   str(new_digit) + state[i+1:])
    30     48.4 MiB      0.0 MiB        818               return res
    31
    32     48.4 MiB      0.0 MiB         10           def bfs(queue: Deque[str], distances: dict,
↪   other_distances: dict) -> int:
    33                                                     """Perform a step of BFS, expanding the
↪   current queue."""
    34     48.4 MiB      0.0 MiB        826               for _ in range(len(queue)):
    35     48.4 MiB      0.0 MiB        818                   current = queue.popleft()
    36     48.4 MiB      0.0 MiB        818                   step_count = distances[current]
    37     48.4 MiB      0.0 MiB       7356                   for nextState in
↪   getNextStates(current):
    38     48.4 MiB      0.0 MiB       6539                       if nextState in deadends_set or
↪   nextState in distances:
    39     48.4 MiB      0.0 MiB       5057                           continue
    40     48.4 MiB      0.0 MiB       1482                       if nextState in other_distances:
    41     48.4 MiB      0.0 MiB          1                           return step_count + 1 +
↪   other_distances[nextState]
    42     48.4 MiB      0.1 MiB       1481                       distances[nextState] = step_count +
↪   1
    43     48.4 MiB      0.0 MiB       1481                       queue.append(nextState)
    44     48.4 MiB      0.0 MiB          8               return -1
    45
    46     48.4 MiB      0.0 MiB          1           deadends_set = set(deadends)
    47                                                 # Early exit if the initial state is a deadend
    48     48.4 MiB      0.0 MiB          1           if '0000' in deadends_set:
    49                                                     return -1
    50
    51                                                 # Initialize BFS structures for bidirectional
↪   search
    52     48.4 MiB      0.0 MiB          1           queue1, queue2 = deque(['0000']),
↪   deque([target])
    53     48.4 MiB      0.0 MiB          1           distance1, distance2 = {"0000": 0}, {target: 0}
    54
    55                                                 # Perform BFS from both ends
    56     48.4 MiB      0.0 MiB          5           while queue1 and queue2:
    57     48.4 MiB      0.0 MiB          5               result = bfs(queue1, distance1, distance2)
    58     48.4 MiB      0.0 MiB          5               if result != -1:
    59                                                         return result
    60     48.4 MiB      0.0 MiB          4               result = bfs(queue2, distance2, distance1)
    61     48.4 MiB      0.0 MiB          4               if result != -1:
    62                                                         return result
    63
    64                                                 return -1
```

Figure 11: A case illustration of GPT-4-turbo-preview (OpenLock). Part G: Memory usage profile.

**Problem:** Given two sorted arrays `nums1` and `nums2` of size `m` and `n` respectively, return **the median** of the two sorted arrays.

The overall run time complexity should be $\mathcal{O}(\log(m + n))$.

**Example 1:**

```
Input: nums1 = [1,3], nums2 = [2]
Output: 2.00000
Explanation: merged array = [1,2,3] and median is 2.
```

**Example 2:**

```
Input: nums1 = [1,2], nums2 = [3,4]
Output: 2.50000
Explanation: merged array = [1,2,3,4]
and median is (2 + 3) / 2 = 2.5.
```

**Constraints:**

- `nums1.length == m`
- `nums2.length == n`
- $0 \leq m \leq 1000$
- $0 \leq n \leq 1000$
- $1 \leq m + n \leq 2000$
- $-10^6 \leq$ `nums1[i]`, `nums2[i]` $\leq 10^6$

```
solution = Solution()
assert solution.findMedianSortedArrays([1, 3], [2]) == 2.0
assert solution.findMedianSortedArrays([1, 2], [3, 4]) == 2.5
```

Figure 12: A case illustration of GPT-4-turbo-preview (FindMedianSortedArrays). Part A: Task Description.

```python
class Solution:
    def findMedianSortedArrays(self, nums1, nums2) -> float:
        A, B = nums1, nums2
        total = len(A) + len(B)
        half = total // 2

        if len(B) < len(A):
            A, B = B, A

        l, r = 0, len(A) - 1
        while True:
            i = (l + r) // 2 # A
            j = half - i - 2 # B

            Aleft = A[i] if i >= 0 else float("-infinity")
            Aright = A[i + 1] if (i + 1) < len(A) else float("infinity")
            Bleft = B[j] if j >= 0 else float("-infinity")
            Bright = B[j + 1] if (j + 1) < len(B) else float("infinity")

            if Aleft <= Bright and Bleft <= Aright:
                if total % 2:
                    return min(Aright, Bright)
                return (max(Aleft, Bleft) + min(Aright, Bright)) / 2
            elif Aleft > Bright:
                r = i - 1
            else:
                l = i + 1
```

Figure 13: A case illustration of GPT-4-turbo-preview (FindMedianSortedArrays). Part B: GPT-4-turbo-preview generated code.

```
Timer unit: 1e-06 s

Total time: 0.00119913 s
File: 4_tmp.py
Function: findMedianSortedArrays at line 45

Line #      Hits         Time  Per Hit   % Time  Line Contents
==============================================================
    45                                               @profile
    46                                               def findMedianSortedArrays(self, nums1, nums2) ->
    ↪  float:
    47       100         41.2      0.4      3.4           A, B = nums1, nums2
    48       100         57.2      0.6      4.8           total = len(A) + len(B)
    49       100         40.6      0.4      3.4           half = total // 2
    50
    51       100         54.9      0.5      4.6           if len(B) < len(A):
    52        46         15.9      0.3      1.3               A, B = B, A
    53
    54       100         42.9      0.4      3.6           l, r = 0, len(A) - 1
    55       169         58.6      0.3      4.9           while True:
    56       169         63.3      0.4      5.3               i = (l + r) // 2 # A
    57       169         62.1      0.4      5.2               j = half - i - 2 # B
    58
    59       169         83.8      0.5      7.0               Aleft = A[i] if i >= 0 else
    ↪  float("-infinity")
    60       169         99.3      0.6      8.3               Aright = A[i + 1] if (i + 1) < len(A)
    ↪  else float("infinity")
    61       169         84.3      0.5      7.0               Bleft = B[j] if j >= 0 else
    ↪  float("-infinity")
    62       169         95.3      0.6      8.0               Bright = B[j + 1] if (j + 1) < len(B)
    ↪  else float("infinity")
    63
    64       169         98.0      0.6      8.2               if Aleft <= Bright and Bleft <= Aright:
    65       100         51.6      0.5      4.3                   if total % 2:
    66        51         86.1      1.7      7.2                       return min(Aright, Bright)
    67        49        103.0      2.1      8.6                   return (max(Aleft, Bleft) +
    ↪  min(Aright, Bright)) / 2
    68        69         36.9      0.5      3.1               elif Aleft > Bright:
    69        45         15.8      0.4      1.3                   r = i - 1
    70                                                       else:
    71        24          8.4      0.3      0.7                   l = i + 1
```

Figure 14: A case illustration of GPT-4-turbo-preview (FindMedianSortedArrays). Part C: Execution time profile.

```
Line #     Mem usage     Increment  Occurrences   Line Contents
==============================================================
    54    24.7 MiB     24.7 MiB           1       @profile(stream=profile_stream,
↪  precision=PROFILE_PRECISION)
    55                                             def findMedianSortedArrays(self, nums1, nums2) ->
↪  float:
    56    24.7 MiB      0.0 MiB           1           A, B = nums1, nums2
    57    24.7 MiB      0.0 MiB           1           total = len(A) + len(B)
    58    24.7 MiB      0.0 MiB           1           half = total // 2
    59
    60    24.7 MiB      0.0 MiB           1           if len(B) < len(A):
    61                                                    A, B = B, A
    62
    63    24.7 MiB      0.0 MiB           1           l, r = 0, len(A) - 1
    64    24.7 MiB      0.0 MiB           1           while True:
    65    24.7 MiB      0.0 MiB           1               i = (l + r) // 2 # A
    66    24.7 MiB      0.0 MiB           1               j = half - i - 2 # B
    67
    68    24.7 MiB      0.0 MiB           1               Aleft = A[i] if i >= 0 else
↪  float("-infinity")
    69    24.7 MiB      0.0 MiB           1               Aright = A[i + 1] if (i + 1) < len(A) else
↪  float("infinity")
    70    24.7 MiB      0.0 MiB           1               Bleft = B[j] if j >= 0 else
↪  float("-infinity")
    71    24.7 MiB      0.0 MiB           1               Bright = B[j + 1] if (j + 1) < len(B) else
↪  float("infinity")
    72
    73    24.7 MiB      0.0 MiB           1               if Aleft <= Bright and Bleft <= Aright:
    74    24.7 MiB      0.0 MiB           1                   if total % 2:
    75                                                            return min(Aright, Bright)
    76                                                        return (max(Aleft, Bleft) + min(Aright,
↪  Bright)) / 2
    77                                                    elif Aleft > Bright:
    78                                                        r = i - 1
    79                                                    else:
    80                                                        l = i + 1
```

Figure 15: A case illustration of GPT-4-turbo-preview (FindMedianSortedArrays). Part D: Memory usage profile.

```python
class Solution:
    def findMedianSortedArrays(self, nums1, nums2) -> float:
        # Ensure nums1 is the smaller array to minimize binary search domain
        if len(nums1) > len(nums2):
            nums1, nums2 = nums2, nums1
        x, y = len(nums1), len(nums2)

        low, high = 0, x
        while low <= high:
            partitionX = (low + high) // 2
            partitionY = (x + y + 1) // 2 - partitionX

            # If partitionX is 0 it means nothing is there on left side. Use -inf for maxLeftX
            maxLeftX = float('-inf') if partitionX == 0 else nums1[partitionX - 1]
            # If partitionX is length of input then there is nothing on right side. Use +inf for
            ↪  minRightX
            minRightX = float('inf') if partitionX == x else nums1[partitionX]

            maxLeftY = float('-inf') if partitionY == 0 else nums2[partitionY - 1]
            minRightY = float('inf') if partitionY == y else nums2[partitionY]

            if maxLeftX <= minRightY and maxLeftY <= minRightX:
                # The partition is correct
                if (x + y) % 2 == 0:
                    return (max(maxLeftX, maxLeftY) + min(minRightX, minRightY)) / 2.0
                else:
                    return max(maxLeftX, maxLeftY)
            elif maxLeftX > minRightY:  # We are too far on right side for partitionX. Go on left
            ↪  side.
                high = partitionX - 1
            else:  # We are too far on left side for partitionX. Go on right side.
                low = partitionX + 1
```

Figure 16: A case illustration of GPT-4-turbo-preview (FindMedianSortedArrays). Part E: Self-Optimization Code.

```
Timer unit: 1e-06 s

Total time: 0.00114754 s
File: 4_tmp.py
Function: findMedianSortedArrays at line 45

Line #      Hits         Time  Per Hit   % Time  Line Contents
==============================================================
    45                                           @profile
    46                                           def findMedianSortedArrays(self, nums1, nums2) ->
   ↪ float:
    47                                               # Ensure nums1 is the smaller array to
   ↪ minimize binary search domain
    48       100         63.3      0.6      5.5       if len(nums1) > len(nums2):
    49        46         18.2      0.4      1.6           nums1, nums2 = nums2, nums1
    50       100         44.7      0.4      3.9       x, y = len(nums1), len(nums2)
    51
    52       100         32.3      0.3      2.8       low, high = 0, x
    53       184         79.6      0.4      6.9       while low <= high:
    54       184         72.1      0.4      6.3           partitionX = (low + high) // 2
    55       184         69.7      0.4      6.1           partitionY = (x + y + 1) // 2 -
   ↪ partitionX
    56
    57                                                   # If partitionX is 0 it means nothing is
   ↪ there on left side. Use -inf for maxLeftX
    58       184         92.6      0.5      8.1           maxLeftX = float('-inf') if partitionX ==
   ↪ 0 else nums1[partitionX - 1]
    59                                                   # If partitionX is length of input then
   ↪ there is nothing on right side. Use +inf for minRightX
    60       184         81.5      0.4      7.1           minRightX = float('inf') if partitionX ==
   ↪ x else nums1[partitionX]
    61
    62       184         94.9      0.5      8.3           maxLeftY = float('-inf') if partitionY ==
   ↪ 0 else nums2[partitionY - 1]
    63       184         86.9      0.5      7.6           minRightY = float('inf') if partitionY ==
   ↪ y else nums2[partitionY]
    64
    65       184        100.1      0.5      8.7           if maxLeftX <= minRightY and maxLeftY <=
   ↪ minRightX:
    66                                                       # The partition is correct
    67       100         52.8      0.5      4.6               if (x + y) % 2 == 0:
    68        49        108.3      2.2      9.4                   return (max(maxLeftX, maxLeftY) +
   ↪ min(minRightX, minRightY)) / 2.0
    69                                                       else:
    70        51         86.7      1.7      7.6                   return max(maxLeftX, maxLeftY)
    71        84         34.7      0.4      3.0           elif maxLeftX > minRightY:  # We are too
   ↪ far on right side for partitionX. Go on left side.
    72        30         11.0      0.4      1.0               high = partitionX - 1
    73                                                   else:  # We are too far on left side for
   ↪ partitionX. Go on right side.
    74        54         18.2      0.3      1.6               low = partitionX + 1
```

Figure 17: A case illustration of GPT-4-turbo-preview (FindMedianSortedArrays). Part F: Execution time profile.

```
Line #     Mem usage     Increment   Occurrences    Line Contents
=============================================================
    54     24.8 MiB     24.8 MiB            1       @profile(stream=profile_stream,
↪    precision=PROFILE_PRECISION)
    55                                              def findMedianSortedArrays(self, nums1, nums2) ->
↪    float:
    56                                                  # Ensure nums1 is the smaller array to minimize
↪    binary search domain
    57     24.8 MiB      0.0 MiB            1           if len(nums1) > len(nums2):
    58                                                      nums1, nums2 = nums2, nums1
    59     24.8 MiB      0.0 MiB            1           x, y = len(nums1), len(nums2)
    60
    61     24.8 MiB      0.0 MiB            1           low, high = 0, x
    62     24.8 MiB      0.0 MiB            1           while low <= high:
    63     24.8 MiB      0.0 MiB            1               partitionX = (low + high) // 2
    64     24.8 MiB      0.0 MiB            1               partitionY = (x + y + 1) // 2 - partitionX
    65
    66                                                      # If partitionX is 0 it means nothing is
↪    there on left side. Use -inf for maxLeftX
    67     24.8 MiB      0.0 MiB            1               maxLeftX = float('-inf') if partitionX == 0
↪    else nums1[partitionX - 1]
    68                                                      # If partitionX is length of input then
↪    there is nothing on right side. Use +inf for minRightX
    69     24.8 MiB      0.0 MiB            1               minRightX = float('inf') if partitionX == x
↪    else nums1[partitionX]
    70
    71     24.8 MiB      0.0 MiB            1               maxLeftY = float('-inf') if partitionY == 0
↪    else nums2[partitionY - 1]
    72     24.8 MiB      0.0 MiB            1               minRightY = float('inf') if partitionY == y
↪    else nums2[partitionY]
    73
    74     24.8 MiB      0.0 MiB            1               if maxLeftX <= minRightY and maxLeftY <=
↪    minRightX:
    75                                                          # The partition is correct
    76     24.8 MiB      0.0 MiB            1                   if (x + y) % 2 == 0:
    77                                                              return (max(maxLeftX, maxLeftY) +
↪    min(minRightX, minRightY)) / 2.0
    78                                                          else:
    79                                                              return max(maxLeftX, maxLeftY)
    80                                                      elif maxLeftX > minRightY:  # We are too
↪    far on right side for partitionX. Go on left side.
    81                                                          high = partitionX - 1
    82                                                      else:  # We are too far on left side for
↪    partitionX. Go on right side.
    83                                                          low = partitionX + 1
```

Figure 18: A case illustration of GPT-4-turbo-preview (FindMedianSortedArrays). Part G: Memory usage profile.

