# OpenReview forum: "EffiLearner: Enhancing Efficiency of Generated Code via Self-Optimization"
_NeurIPS.cc/2024/Conference — NeurIPS 2024 poster_

### Official Review · Reviewer_pc3i · 2024-07-01

**Soundness:** 1
**Presentation:** 4
**Contribution:** 2
**Rating:** 4
**Confidence:** 5

**Summary:**

The paper aims to address the inefficiency issue of code generated by LLMs in terms of execution time and memory consumption. The authors propose a framework called Self-Optimization based on OverheAd Profile (SOAP), which improves the efficiency of LLM-generated code by iteratively refining it based on execution overhead profiles. These profiles capture the code's execution time and memory usage, which are then fed back into the LLM to optimize the code. The paper demonstrates the effectiveness of SOAP through extensive experiments on various benchmarks and selected LLMs. Results show reductions in execution time and memory usage, highlighting SOAP's ability to enhance code efficiency without compromising correctness.

**Strengths:**

* The paper is well presented and the targeted problem of code inefficiency is well motivated.
* The evaluation covers a good number of Python coding tasks and LLMs.
* The authors mentioned that the code of the work will be open-sourced.

**Weaknesses:**

* The baseline selection can be largely improved. The current baseline in the paper is to ask LLMs to directly generate code -- it is not the models' fault to generate non-optimal code when they are not asked to. At least two reasonable and simple baselines should have been included: (1) ask the model to generate an efficient version of code, and (2) ask the model to generate code and then directly optimize it (similar to the ALGO approach, despite not using test generation and execution to soften the environmental assumptions). Furthermore, the PIE [1] paper mentioned various few-shot prompting methods to perform code optimization, which should also be used as baselines.
* To perform an "apple-to-apple" comparison over the models, Tables 1-4 should be better aligned. To understand which model is better at generating efficient code, they must be compared under the same set of tasks. While due to correctness inconsistency, the attributed coding tasks are not aligned, authors may compare the models under a set of mutually passing coding tasks.
* Efficiency evaluation can be noisy without carefully configuring the metrics and testbeds. The considered metrics such as execution time and memory usage are flaky and non-reproducible as they are measured in physical platforms. To mitigate these limitations, authors may follow PIE [1] to use architecture simulators or other better-defined metrics.
* While the author highlighted efficiency is critical in resource-constrained scenarios, the proposed technique is specialized for Python. It is unclear if Self-Optimization can work for performance-demanding languages such as C/C++ and related experiments are ignored.
* [1] is a closely relevant and published work that should be well discussed and compared.

[1] Shypula A, Madaan A, Zeng Y, Alon U, Gardner J, Hashemi M, Neubig G, Ranganathan P, Bastani O, Yazdanbakhsh A. “Learning Performance-Improving Code Edits.” ICLR 2024.

**Questions:**

* How is SOAP compared to more reasonable baselines below:
  1. Prompting an LLM to generate efficient code in zero-shot
  2. Ask the LLM to refine its first generation for code performance
  3. Fewshot prompting mentioned in PIE
* Are profilers necessary? What if we just replace profilers with self-reasoning or self-reflection?
* How's the cost (in the form of token consumption and pipeline running time) of SOAP and baselines?
* How is the overall variation of mentioned metrics including execution time? For example, when running and measuring the code execution for N times, what is the overall coefficient of variation?

**Limitations:**

* Using a profiler and code executor in the pipeline is expensive with respect to the degree of improvement. In production, integrating code execution in a sandbox is already challenging, and integrating a profiler (for multiple languages) would take more engineering cost. Meanwhile, the performance gain (e.g., execution time) overall looks minimal compared to the baseline whose prompt does not even encourage code optimization.
* Overall the technical contribution of the paper is incremental: compared to prior techniques such as ALGO and PIE, the major difference is the use of code profiler.

---

> ### Author Rebuttal · Authors · 2024-08-06
>
> Thank you for your thoughtful review and insightful comments. We hereby address your concerns below:
>
> **W1 & Q1 & W5: Baseline selection can be largely improved.**
>
> Thanks for your suggestion. We provide the evaluation results of SOAP and suggested baselines in **Author Rebuttal Table 1**.
>
> - For the first suggested method, we named it **Directly Efficiency**, where LLMs are required to generate efficient code for the task description.
>
> - For the second suggested method, we name it **Self-Refine Efficiency**, where we ask LLMs to generate code and then directly optimize it.
> Then, since PIE fine-tunes LLMs to improve the efficiency of LLM-generated code, which is orthogonal with SOAP, we conduct experiments for PIE and PIE + SOAP for five different prompts to demonstrate that with SOAP, PIE fine-tuned LLMs generated code could be further improved.
>
> As shown in **Author Rebuttal Table 1**, we can observe that SOAP achieves SOTA results compared with Directly Efficiency and Self-Refine Efficiency. For example, Directly Efficiency and Self-Refine Efficiency decreases NTMU from 19.65 to 2.88 and 2.65, while SOAP decreases from 19.65 to 1.64. Besides, we can observe that with SOAP, the efficiency of PIE-generated code would further improve, e.g., decreasing the average execution time from 0.74s to 0.41s for PIE+Dynamic Retrieval,k=4.
>
>
> **W2: Compare the models under a set of mutually passing coding tasks.**
>
> Thanks for your suggestion. The evaluation results are shown in **Author Rebuttal Table 2**, where we conduct experiments on seven closed-source LLMs with 210 tasks, which have higher pass@1 compared with open-source LLMs, and then can collect massive tasks addressed by all closed-source LLMs. As shown in Table 8, we can observe that SOAP with GPT-3.5-turbo-0301 achieves SOTA efficiency compared with other LLMs. For example, the average execution time decreases from 0.37s to 0.28s.
>
>
> **W3: Follow PIE to use architecture simulators or other better-defined metrics.**
>
> We provide the evaluation results of SOAP and the initial code in **Author Rebuttal Table 3**. We can observe that in the PIE-provided default simulator, SOAP can also improve the efficiency of LLM-generated initial code. For example, the average execution time of OpenCodeInterpreter-1.3B decreases from 0.80s to 0.52s.
>
>
> **W4: Unclear if Self-Optimization can work for performance-demanding languages such as C/C++.**
> To address the reviewer's concern, we conduct experiments for SOAP in the HumanEval-ET (C++) dataset. Our evaluation results are shown below, where we can observe that **SOAP can also improve the efficiency of LLM-generated C++ programs**. For example, the average execution time of CodeLlama-70B-Instruct-hf decreases from 667.9ms to 459.5ms.
>
> | Evaluation in HumanEval (C++)     | ET (ms) | NET  | MU (KB) | NMU  | TMU (KB*ms) | NTMU |
> |------------------|--------|------|---------|------|------------|------|
> | CodeLlama-70b-Instruct-hf       |        |      |         |      |            |      |
> | Initial Version                 | 667.9  | 1.5  | 93.2    | 1.3  | 58.9       | 2.1  |
> | SOAP                            | 459.5  | 1.0  | 72.2    | 1.0  | 34.1       | 1.3  |
> | gpt-3.5-turbo-0301              |        |      |         |      |            |      |
> | Initial Version                 | 668.1  | 1.5  | 78.9    | 1.1  | 79.0       | 2.6  |
> | SOAP                            | 577.3  | 1.2  | 71.5    | 1.0  | 63.8       | 2.1  |
>
> **Q2: Are profilers necessary? What if we just replace profilers with self-reasoning or self-reflection?**
>
>
> Profilers are necessary for the code efficiency optimization process. Specifically, to demonstrate the importance of profilers, we conduct experiments for straightforward code optimization (Unsupervised Self-Refine and Result-Aware Self-Refine), reasoning-based code optimization (self-reasoning and self-reflection), single profiler-based code optimization, and provide both memory profiler and execution time profiler (SOAP).
>
> The evaluation results are shown below. Our experiments demonstrate that by providing both memory profiler and execution time profiler (SOAP), the code efficiency achieves SOTA results compared with other methods. For example,  SOAP decreases the average execution time from 0.36s to 0.28s.
>
>
> |OptimizationProfile|ET(s)|NET|MU(Mb)|NMU|TMU(Mb*s)|NTMU|
> |---|---|---|---|---|---|---|
> |**GPT-3.5-Turbo-0301**|||||||
> |InitialVersion|0.36|2.50|91.25|2.45|157.50|19.75|
> |Unsupervised Self-Refine|0.32|2.46|78.39|2.12|312.99|42.42|
> |Result-Aware Self-Refine|0.30|2.25|58.65|1.61|195.49|27.16|
> |Self-Reasoning|0.89|6.21|60.64|1.62|45.91|5.61|
> |Self-Relfection|0.81|5.67|60.64|1.62|39.35|4.80|
> |SOAP|0.28|2.01|36.08|0.99|12.43|1.64|
>
>
> **Q3: How's the cost (token consumption and running time) of SOAP and baselines?**
>
> To address the reviewer’s concern about the cost of SOAP and baselines, we provide the evaluation results in **Author Rebuttal Table 4**, where we can observe that the initially generated code requires 1.1M token usage and 76.81s for all tasks. While SOAP requires 5.4M token usage and 566.67s for all tasks. Although SOAP requires more tokens compared with the initial version, the execution time and memory usage (See paper table 1) are largely improved. We believe that this exact token usage is affordable.
>
>
> **Q4: When running and measuring the code execution for N times, what is the overall coefficient of variation?**
>
> The overall coefficient of variation (CoV) for each model and metric in the SOAP benchmark is presented in **Author Rebuttal Table 5**, which is calculated by running and measuring the code execution 5 times for each model, then computing the mean and standard deviation of each metric. From the table, we can observe that all of the metrics' coefficients of variation are lower than 3% for all models, suggesting that the **performance of these models is relatively consistent across multiple runs**.

---

> > ### Comment · Reviewer_pc3i · 2024-08-11
> > **Thanks for the detailed reply!**
> >
> > First, I'd like to thank the authors for their reply. It is impressive and surprising to add this volume of preliminary experiments within such a short period.
> >
> > While I think weaknesses #1, #4, and #5 can be mostly addressed by adding new tables in the paper, weaknesses #2 and #3 require a more thorough global revision/update. Specifically, while in the demonstrated comparison from the preliminary results SOAP outperforms other baselines, the ratio of improvements is largely different from table to table when using different measurements and task set alignments. For example, improvements in Table 3 are way larger than that of Table 2 and this is likely due to the impact of task correctness over efficiency.
> >
> > That being said, while I appreciate the various new preliminary experiments from the authors, I still want to hold a rejection before an optimal experimental setting is figured out and applied globally. This would largely improve conclusion consistency and soundness of evaluation, which is very crucial for efficiency evaluation.
> >
> > Regarding the responses to my questions, I have a few concerns:
> > 1. The variations in Table 5 are surprisingly small and the variations across different metrics are surprisingly equivalent. Can the authors double-check their data?
> > 2. How do the authors configure Result-Aware Self-Refine compared to SOAP?
> >
> > Follow-up suggestions:
> > 1. Given the large token overhead of SOAP, what if we just do optimization sampling and profile to select the best one using the same token budget?
> > 2. In general, it is encouraged to simplify the metrics for clarity.

---

> > > ### Author Response · Authors · 2024-08-11
> > > **Rebuttal for Reviewer 4 Follow Up**
> > >
> > > **Specifically, in the demonstrated comparison from the preliminary results SOAP outperforms other baselines, the ratio of improvements is largely different from table to table when using different measurements and task set alignments. For example, improvements in Table 3 are way larger than that of Table 2 and this is likely due to the impact of task correctness over efficiency. That being said, while I appreciate the various new preliminary experiments from the authors, I still want to hold a rejection before an optimal experimental setting is figured out and applied globally. This would largely improve the consistency and soundness of the conclusion, which is crucial for efficiency evaluation.**
> > >
> > >
> > > Response:
> > >
> > > Thank you for your appreciation of the new preliminary experiments we provided during the rebuttal process. However, we are disheartened that reviewer pc3i wishes to reject our paper based on the claim, **"I still want to hold a rejection before an optimal experimental setting is figured out and applied globally"**. To address reviewer pc3i's concerns about w2 and w3, we have provided new results to reviewer pc3i's requirements. Specifically, for w2, we want to clarify that we conducted experiments on **all** closed-source models. However, **we did not conduct experiments on open-source models due to their low pass@1 and the lack of consistent correct code across all open-source models**.
> > >
> > >
> > > Next, for the PIE provided Gem5 default simulator, we also included seven different LLMs: smaller open-source LLMs (OpenCodeInterPreter-1.3B, DeepSeekCoder-1.3B-Ins, CodeLlama-7B-Ins-hf, StarCoder2-15B), a larger open-source model (CodeLlama-70B-Ins-hf), and two closed-source models (GPT-3.5-turbo-0301 and Claude-3-sonnet). We believe that these models are representative of our experiments.
> > >
> > >
> > >
> > >
> > > **The variations in Table 5?**
> > >
> > >
> > > Response:
> > >
> > > To address the reviewer’s concern, we provide the source code in our anonymous GitHub. We first use run_source_code_five_times.py to run the LLM-generated code five times. Next, we use report_variant.py to first calculate mean and std. Then we report coef_of_var with (std_dev / mean) * 100 for all experiments.
> > >
> > >
> > > **Configuration of Result-Aware Self-Refine compared to SOAP?**
> > >
> > > Response:
> > >
> > > The configuration of Result-Aware Self-Refine is shown in our anonymous GitHub prompts.py, where the only difference between SOAP and Result-Aware Self-Refine is that the overhead analysis part of Result-Aware Self-Refine in Lines 19 only contains the ET, MU, and TMU (See Lines 54-58). The overhead analysis part of SOAP also contains the execution time profiler and memory profile (See Lines 28-35).
> > >
> > >
> > > **Follow-up suggestions:**
> > >
> > > **S1: Given the large token overhead of SOAP, what if we just do optimization sampling and profile to select the best one using the same token budget?**
> > >
> > > Response:
> > >
> > > Thank you for your suggestion. We have provided evaluation results for your suggested method and SOAP in GPT-3.5-turbo-0301 below. It's observable that this method can enhance the efficiency of code generated by the LLM as compared to the initial version. For instance, the TMU of GPT-3.5-turbo-0301 decreases from 157.50 (Mbs) to 27.99 (Mbs). This significant reduction is primarily due to the suggested method of selecting the most efficient route from multiple codes. However, we also noticed that the suggested method is still less efficient than SOAP. The primary reason for this is that the suggested method primarily aims to generate multiple efficient solutions for a given task. However, without the profile information provided by SOAP, the LLM is unable to optimize the code and generate a more efficient version than the initial code, resulting in the suggested method being less efficient than SOAP.
> > >
> > > | Strategy | **ET (s)** | **NET** | **MU (Mb)** | **NMU** | **TMU (Mb*s)** | **NTMU** |
> > > |----------|------------|---------|-------------|---------|----------------|----------|
> > > |Initial|0.36|2.50|91.25|2.45|157.50|19.75|
> > > | Reviewer pc3i suggested | 0.31 | 2.17 | 36.14 | 1.00 | 27.99 | 2.94 |
> > > | SOAP | 0.28 | 2.01 | 36.08 | 0.99 | 12.43 | 1.64 |
> > >
> > >
> > > *Effectiveness of SOAP with Reviewer pc3i proposed optimization sampling and profile to select the best one for GPT-3.5-turbo with similar token usages. For each method, we use **int(token usage of SOAP/token usage of baselines such as direct generation with sampling) + 1** to ensure that the token usage of SOAP should be lower than the baselines.*
> > >
> > > **S2: In general, it is encouraged to simplify the metrics for clarity.**
> > >
> > > Response:
> > >
> > > Thanks for your suggestion. We will revise our manuscript in the final version.

---

> ### Comment · Reviewer_pc3i · 2024-08-11
> **Clarification on rejection**
>
> Thanks for the reply. I am sorry for not explaining my decision enough and here are my detailed reasons for my suggestion of decision:
>
> 1. First, the preliminary experimental results from the authors quickly cleared my concerns about approach effectiveness regarding weaknesses in #1, #4, and #5.
> 2. I won't reject the paper just for #1, #4, and #5 because those results are complementary to what the paper already had, meaning that they could be easily added to the camera-ready version if we end up accepting it.
> 3. Yet, weakness #2 still holds as existing results of the paper are mostly not represented in an "apple-to-apple" fashion. It is not even right to compare the efficiency of LLM codegen over different sets of tasks, making the efficiency score unreasonable to compare. For an extreme case, two LLMs may solve completely different sets of solutions, and computing efficiency scores over two totally different task sets cannot lead to clean and reasonable results regarding code efficiency in general.
> 4. Because it is problematic to compare LLM efficiency over different sets of tasks, ideally the author should globally correct the evaluation methodology, i.e., removing results made by inconsistent comparison and enforcing correctness alignments. This is crucial before accepting this paper because if not future work might inherit such a misevaluation methodology, leading to more inconsistent comparison in general.
> 5. Yet, doing reason 4 in rebuttal is quite hard objectively speaking. Meanwhile, given it's a major revision, it might be more rigorous to apply and reorganize the experiments and resubmit a clean version for another review, rather than accepting the paper through some rough views over the set of preliminary results that we are unsure if those can be systematically adopted in the camera-ready.
>
> To summarize, I deeply appreciate the efforts of the authors in the rebuttal; however, for soundness, the required revision goes beyond "convincing the reviewers through proof-of-concept experiments" that it is crucial to globally correct the evaluation methodology for positively leading future research in this thread. Last but not least, I strongly feel this work can be accepted after doing a major revision.

---

> ### Author Response · Authors · 2024-08-12
> **Apple-to-Apple results**
>
> Thank you for your reply. SOAP is effective for both scenarios (i.e., improving LLMs' own tasks and improving common correct tasks).
>
> To address the concern of the "apple-to-apple" results of our experiments in the Paper's main flow, in this thread, we provide results based on common tasks for Table 2 - Table 4's results, which provide consistent results for the two configurations.
> (*Table 1's results are already demonstrated in a previous thread.*)
>
> For Paper Table 2, where we discuss how SOAP's effectiveness would be affected by the optimization results, we provide the evaluation results in **Reviewer pc3i Table 2 (common tasks)**. First, we find the tasks addressed by both CodeLlama-70B-Ins and GPT-3.5-turbo-0301 with the Initial version (Step 0) and SOAP optimization process (Step 1-5). Then, we collect 46 tasks and measure the efficiency of LLM-generated code in these tasks. The evaluation results in **Reviewer pc3i Table 2 (common tasks)** show that CodeLlama-70B-Ins achieves SOAP efficiency results with one-step SOAP optimization, while GPT-3.5-turbo-0301 continuously improves the efficiency for the evaluated 46 tasks from Step 0 to Step 3. Then, the efficiency of GPT-3.5-turbo-0301 maintains efficiency for the other two optimization steps.
>
>
> Next, for Paper Table 3, where we discuss how different prompt optimization steps affect the efficiency of LLM-generated code, we provide the evaluation results in **Reviewer pc3i Table 3 (common tasks)**. First, we find the tasks addressed by both CodeLlama-70B-Ins and GPT-3.5-turbo-0301 for six different prompts and then get 40 tasks. Then, we provide the efficiency results in **Reviewer pc3i Table 3 (common tasks)**, where we can observe that SOAP also achieves SOTA efficiency results compared with other baselines. For example, SOAP achieves 3.72 (Mb*s) for TMU, while baselines only achieve 3.84 (Mb*s) in the TMU metric for GPT-3.5-turbo generated code.
>
>
> Finally, for Paper Table 4, where we discuss the differences in the efficiency of SOAP and the initial version of the CodeLlama family, we provide the evaluation results in **Reviewer pc3i Table 4 (common tasks)**. First, we detect correct tasks addressed by both SOAP and the Initial version for four CodeLlama models and then get 3 tasks. Next, we provide the efficiency results of CodeLlama family-generated code in **Reviewer pc3i Table 4 (common tasks)**. We can observe that for four CodeLlama models, SOAP continuously improves the efficiency of LLM-generated code.
>
>
> We hope that our newly provided results can address the reviewer's concern.
>
> *Note: We provide the Tables in the next thread.*

---

> ### Author Response · Authors · 2024-08-12
> **Results**
>
> ***Reviewer pc3i Table 2 (common tasks)** Effect of the number of self-optimization steps in SOAP (46 tasks). The evaluation results are used to replace the results of Paper Table 2 with the same correct tasks addressed by two LLMs for all steps.*
>
>
> | Steps | **ET (s)** | **NET** | **MU (Mb)** | **NMU** | **TMU (Mb*s)** | **NTMU** |
> |-------|------------|---------|-------------|---------|-------|----------|
> |||  | |  |  ||
> | **CodeLlama-70b-Instruct-hf** ||  | |  |  ||
> | 0| 0.40| 3.00 | 54.79| 1.76 | 20.05| 4.86|
> | 1| 0.29| 2.20 | 54.90| 1.77 | 14.58| 3.53|
> | 2| 0.29| 2.20 | 54.90| 1.77 | 14.58| 3.53|
> | 3| 0.29| 2.20 | 54.90| 1.77 | 14.58| 3.53|
> | 4| 0.29| 2.20 | 54.90| 1.77 | 14.58| 3.53|
> | 5| 0.29| 2.20 | 54.90| 1.77 | 14.58| 3.53|
> |||  | |  |  ||
> | **gpt-3.5-turbo-0301** ||  | |  |  ||
> | 0| 0.24| 1.78 | 31.12| 1.00 | 10.74| 2.60|
> | 1| 0.22| 1.66 | 31.28| 1.01 | 7.36 | 1.78|
> | 2| 0.22| 1.64 | 31.27| 1.01 | 7.02 | 1.70|
> | 3| 0.22| 1.64 | 31.27| 1.01 | 6.98 | 1.69|
> | 4| 0.22| 1.64 | 31.27| 1.01 | 6.98 | 1.69|
> | 5| 0.22| 1.64 | 31.27| 1.01 | 6.98 | 1.69|
>
>
>
>
>
>
> ***Reviewer pc3i Table 3 (common tasks)** Effect of the number of self-optimization steps in SOAP (40 tasks). The evaluation results are used to replace the results of Paper Table 3 with the same correct tasks addressed by two LLMs for all steps.*
>
>
> | Steps| ET (s) | NET| MU (Mb) | NMU | TMU (Mb*s) | NTMU |
> | ------------------------- | ------ | ----- | ------- | --- | ---------- | ---- |
> | **CodeLlama-70b-Instruct-hf**  | ||  |  |  ||
> | Initial  | 1.82| 14.78 | 25.85| 1.00| 36.32| 16.89|
> | Unsupervised Self-Refine  | 0.34| 2.80  | 25.82| 1.00| 6.99| 3.25 |
> | Result-Aware Self-Refine  | 0.33| 2.67  | 25.96| 1.00| 6.85| 3.19 |
> | memory_profiler | 0.26| 2.08  | 25.88| 1.00| 4.38| 1.97 |
> | time_overhead_profiler | 0.19| 1.57  | 25.88| 1.00| 4.03| 1.89 |
> | SOAP | 0.19| 1.55  | 25.87| 1.00| 3.98| 1.85 |
> | **gpt-3.5-turbo-0301** | ||  |  |  ||
> | Initial  | 0.24| 1.95  | 25.91| 1.00| 5.01| 2.33 |
> | Unsupervised Self-Refine  | 0.26| 2.14  | 29.99| 1.16| 7.65| 3.56 |
> | Result-Aware Self-Refine  | 0.23| 1.87  | 25.95| 1.00| 5.12| 2.38 |
> | memory_profiler | 0.19| 1.53  | 25.88| 1.00| 3.84| 1.79 |
> | time_overhead_profiler | 0.18| 1.47  | 31.26| 1.21| 4.98| 2.32 |
> | SOAP | 0.18| 1.48  | 25.88| 1.00| 3.72| 1.73 |
>
>
> ***Reviewer pc3i Table 4 (common tasks)** Effect of the number of self-optimization steps in SOAP (3 tasks) for CodeLlama family.*
>
> | Steps | **ET (s)** | **NET** | **MU (Mb)** | **NMU** | **TMU (Mb*s)** | **NTMU** |
> |-------|------------|---------|-------------|---------|-------|-----|
> ||   ||||| |
> | **CodeLlama-7b-Instruct-hf** |   ||||| |
> | Initial | 0.78 | 5.61 | 30.82 | 1.00 | 13.89 | 5.50 |
> | SOAP (step 5) | 0.23 | 1.65 | 54.69 | 1.77 | 7.80 | 3.09 |
> ||   ||||| |
> | **CodeLlama-13b-Instruct-hf** |   ||||| |
> | Initial | 0.33 | 2.38 | 54.66 | 1.76 | 11.44 | 4.53 |
> | SOAP (step 5) | 0.25 | 1.76 | 54.69 | 1.77 | 8.46 | 3.35 |
> ||   ||||| |
> | **CodeLlama-34b-Instruct-hf** |   ||||| |
> | Initial | 0.31 | 2.25 | 54.59 | 1.76 | 10.72 | 4.25 |
> | SOAP (step 5) | 0.25 | 1.76 | 54.64 | 1.76 | 8.42 | 3.33 |
> ||   ||||| |
> | **CodeLlama-70b-Instruct-hf** |   ||||| |
> | Initial | 0.32 | 2.31 | 54.62 | 1.76 | 10.97 | 4.34 |
> | SOAP (step 5) | 0.23 | 1.61 | 54.71 | 1.77 | 7.65 | 3.03 |

---

> > ### Author Response · Authors · 2024-08-14
> > **Friendly Reminder: Pending Reviewer Responses and Assessment Consideration**
> >
> > Dear Reviewer pc3i,
> >
> > Thank you for your review and comments on our paper, which you provided on August 12th.
> >
> > To address your concerns regarding the "apples to apples" comparison, we have added all experiments in the last response thread with the same format for our paper. We believe this will facilitate a more accurate and fair evaluation of our work.
> >
> > We would greatly appreciate it if you could take the time to review these results and consider improving your overall assessment of our paper based on these revisions.
> >
> > Thank you once again for your valuable feedback and for considering our request.

---

### Official Review · Reviewer_VC11 · 2024-07-05

**Soundness:** 3
**Presentation:** 4
**Contribution:** 4
**Rating:** 8
**Confidence:** 5

**Summary:**

This paper studies an important and timely issue: the inefficiency often found in code generated by current Large Language Models (LLMs), which can result in longer execution times and higher memory consumption. To mitigate this issue, the paper proposes a self-optimization framework SOAP, designed to improve the efficiency of LLM-generated code. SOAP starts from the code generated by an LLM and then executes it locally to profile its execution time and memory usage. These profiles are fed back to the LLM, allowing it to revise the code and reduce overhead iteratively. The authors verify the effectiveness of SOAP through empirical experiments on the leading open-source and closed-source models, showing SOAP can substantially optimize both execution time and memory consumption.

**Strengths:**

1) This paper proposes a novel general framework to optimize the efficiency of code generated by LLMs. It utilizes execution overhead profiles for self-diagnosis and iteratively enhances code efficiency.

2) Empirical experiments conducted on various leading LLMs demonstrate that this framework effectively optimizes both the execution time and memory consumption of LLM-generated code.

3) The framework does not require additional model training or tuning, making it flexible enough to be attached to arbitrary LLMs as a standalone pipeline.

4) The paper is well-written; the methodology introduced is concise, and the experimental results are robust and clear.

**Weaknesses:**

1) The multi-iteration optimization method requires large token overhead, which can be both resource-intensive and time-consuming.

2) Overhead profiles serve as a good indicator for local code optimization. Nevertheless, the framework may struggle with the scalability of extended code global optimization due to the limited context window of LLMs.

**Questions:**

1) In this paper, SOAP optimizes both execution time and memory usage simultaneously. However, in real-world scenarios, people might prioritize a single objective, either time or space. Do you think it is possible for SOAP to include optimization options (similar to -O3 or -Oz in GCC) to cater to different preferences?

2) The iterative self-optimization process might increase the risk that the final generated code is either non-executable or deviates from the original problem specifications. Do you have any mechanisms in place to prevent such deterioration?

3) How does the token usage in the SOAP profile optimization process compare to that in direct code generation (one-shot)?

**Limitations:**

The authors addressed the limitations and potential negative societal impact of their work.

---

> ### Author Rebuttal · Authors · 2024-08-06
>
> Thank you for your thoughtful review and insightful comments. We hereby address your concerns below:
>
>
> **W1 & W2 & Q3: The multi-iteration optimization method requires a large token overhead.**
>
>
> To address the reviewer's concern about overhead and context window limitations, we provide detailed metrics on execution time, token usage, per iteration input/output token usage， and efficiency in the Table below:
>
>
> | Model | Execution Time (s) | Token Usage (million) | Per Iteration Input/Output Task Token Usage (K) |ET(s)|NET|MU(Mb)|NMU|TMU(Mb*s)|NTMU|
> |-------|-------------------|----------------------|--------------------------------------------------|---|---|---|---|---|---|
> |InitialVersion|76.81|1.1|1.1|0.36|2.50|91.25|2.45|157.50|19.75|
> |UnsupervisedSelf-Refine|416.81|3.9|1.16|0.32|2.46|78.39|2.12|312.99|42.42|
> |Result-AwareSelf-Refine|419.24|3.9|1.16|0.30|2.25|58.65|1.61|195.49|27.16|
> |LineProfiler|555.68|4.7|1.49|0.33 |2.34 |36.43 |0.99|14.07 |1.81|
> |MemoryProfiler|536.26|4.7|1.49|0.34 |2.40 |36.85 |1.00 |16.34 |2.10|
> |SOAP|566.67|5.4|1.78|0.28|2.01|36.08|0.99|12.43|1.64|
>
>
> *Table: Overhead of different code efficiency optimization methods for GPT-3.5-turbo.*
>
>
> We can observe the following:
>
>
> - **Execution Time and Token Overhead**: SOAP requires approximately 8x more execution time compared to the Initial Version. However, this overhead is justified as the optimized code resulting from SOAP significantly reduces the execution time and memory usage for real-world software that could be executed millions of times. Besides, while the optimization process itself is resource-intensive, it yields substantial efficiency gains in deployment scenarios. For example, the average memory peak (MU) of SOAP-generated code **only requires 40%** compared with the initially generated code, which can help source code applied in memory-constrained environments, such as embedded systems or mobile devices. Furthermore, the reduced memory footprint and improved execution speed of the optimized code can lead to better overall system performance, especially in scenarios where the software is frequently used or runs on resource-limited hardware. As a result, the upfront computational cost of the optimization process is offset by the long-term benefits of more efficient and lightweight code in real-world applications.
>
> - **Context Window and Scalability**: SOAP requires **an average of 1.78K tokens** for each input+output task iteration. Given that existing LLMs such as GPT-3.5 and GPT-4 have context windows of 4K tokens or more, they can effectively handle the global profiler information provided by SOAP. This capability allows the LLMs to understand comprehensive profiling data and perform global code optimizations, addressing concerns about scalability within the context window limitations.
>
>
> By leveraging the available context window efficiently and optimizing the code based on detailed profiling data, SOAP manages to enhance the performance and scalability of the generated code, making it a valuable tool despite its initial overhead.
>
>
> **Q1: Include optimization options (similar to -O3 or -Oz in GCC) to cater to different preferences.**
>
>
> Thanks for your suggestion. As shown in Table 3 in the paper, the memory profiler and line profiler focus on the memory usage and execution time individually. To specify the optimization options, developers can directly revise Lines 508-509 in the [Link](https://anonymous.4open.science/r/SOAP-FF6C/src/code_efficiency_calculator.py) to specify the optimization object (e.g., execution time and memory usage). In our final version, we will refine our source code argparse to provide optimization options for SOAP in the version.
>
>
> **Q2: Mechanisms to prevent code deterioration?**
>
>
> To mitigate this risk and ensure the reliability of our experiments, we have implemented a robust two-step verification process:
>
>
> - **Initial Verification**: During the optimization process, if the refined code fails to pass the public test cases, we revert to using the initial version of the LLM-generated code for that specific task. This ensures that any code considered in our experiments at least meets the baseline correctness as defined by the public test cases.
>
>
> - **Comprehensive Filtering**: In cases where the code passes the public test cases but fails the private test cases, we exclude the entire task from our experiments for that respective model. Consequently, we do not calculate the optimized code's efficiency for any step of that task. For example, if the code optimized in step 3 is correct but the code in step 4 is incorrect for a specific task, we exclude all steps of that task from our efficiency calculations for the corresponding model.
>
>
> By adhering to this two-step process, we ensure that only fully verified and correct code is included in our evaluations. This approach allows us to provide a fair and accurate comparison of code efficiency across different optimization steps and models, thereby preventing any deterioration in code quality through the iterative optimization process.

---

> > ### Comment · Reviewer_VC11 · 2024-08-11
> > **Thank you for your response**
> >
> > Thank you for your thorough responses. The additional experiments and explanations have satisfactorily addressed my concerns. As a result, I am pleased to elevate my confidence from 4 to 5 and advocate for the paper's acceptance.
> >
> > I also read comments from the Area Chair and other Reviewers. I like the idea of evaluating N code variants x M LLMs, as it promises to provide a comprehensive performance spectrum across various LLMs. However, it is clear that conducting such an intensive experiment within the rebuttal period is impractical. Could you share the individual model performances (or percentages) from your experiment in the AC thread? It would be intriguing to see the expertise demonstrated by each individual model.

---

> > > ### Author Response · Authors · 2024-08-12
> > >
> > > Thanks for your reply. We provide the distribution below. We can observe that GPT-4-turbo-preview has the highest distribution in Pre-SOAP efficiency results. GPT-3.5-turbo-0301 achieves the highest efficiency ratio in Post-SOAP efficiency results.
> > >
> > >
> > > *Distribution of the most efficient code for each task of EffiBench. We provide each LLM ratio for the most efficient code and then report the ratio for the overall results.*
> > >
> > >
> > > | Model                             | Pre-SOAP (%)| Post-SOAP (%)|
> > > |-----------------------------------|----------|-----------|
> > > | OpenCodeInterpreter-DS-1.3B       | 2.33     | 1.78      |
> > > | OpenCodeInterpreter-DS-6.7B       | 4.66     | 2.88      |
> > > | OpenCodeInterpreter-DS-33B        | 6.30     | 3.70      |
> > > | deepseek-coder-1.3b-instruct      | 3.84     | 1.10      |
> > > | deepseek-coder-6.7b-instruct      | 5.07     | 7.81      |
> > > | deepseek-coder-33b-instruct       | 0.00     | 0.00      |
> > > | CodeLlama-7b-Instruct-hf          | 1.23     | 0.55      |
> > > | CodeLlama-13b-Instruct-hf         | 3.29     | 0.82      |
> > > | CodeLlama-34b-Instruct-hf         | 4.52     | 1.51      |
> > > | CodeLlama-70b-Instruct-hf         | 5.34     | 0.82      |
> > > | XwinCoder-13B                     | 4.11     | 0.55      |
> > > | XwinCoder-34B                     | 5.75     | 2.74      |
> > > | WizardCoder-Python-13B-V1.0-GPTQ  | 1.10     | 0.41      |
> > > | starcoder2-3b                     | 0.68     | 1.23      |
> > > | starcoder2-7b                     | 1.37     | 0.41      |
> > > | starcoder2-15b                    | 0.55     | 0.14      |
> > > | gpt-3.5-turbo-0301                | 8.77     | 27.67     |
> > > | gpt-3.5-turbo-0613                | 7.53     | 5.07      |
> > > | gpt-3.5-turbo-1106                | 7.12     | 4.52      |
> > > | gpt-4                             | 7.53     | 7.53      |
> > > | gpt-4-turbo-preview               | 16.16    | 14.25     |
> > > | claude-3-haiku                    | 1.37     | 5.89      |
> > > | claude-3-sonnet                   | 1.37     | 8.63      |

---

### Official Review · Reviewer_5c1j · 2024-07-08

**Soundness:** 3
**Presentation:** 2
**Contribution:** 3
**Rating:** 6
**Confidence:** 4

**Summary:**

The paper introduces a novel method for code super-optimization by iteratively refining code using LLMs with profiling information. The focus is on enhancing the efficiency of LLM-generated code, addressing execution time and memory requirements. The proposed methodology involves generating code with an LLM and then iteratively feeding profiling results back to the LLM to optimize the code's efficiency. The profiling information includes per-line execution time and memory usage data, helping to identify specific lines that require significant overhead. Experimental results show that this approach outperforms existing unsupervised self-refine and result-aware self-refine methods on efficiency benchmarks like EffiBench, with supplementary experiments on HumanEval and MBPP.

**Strengths:**

- The paper addresses a gap in current LLM code generation research by focusing on code efficiency.
- The proposed iterative refinement method with profiling information is still innovative and provides a clear path to optimizing execution time and memory usage.
- Experimental results demonstrate that the method significantly outperforms existing approaches in terms of code efficiency, showing the potential practical value of this technique.

**Weaknesses:**

- LLM self-refinement or self-repair techniques have been extensively employed in generating responses/codes [1] and debugging tasks [2], whereas utilizing code profiling information for optimization is also a long-established concept [3, 4], which raises concerns in the proposed method's technical novelty.
- The paper seems to treat the number of optimization steps as a hyperparameter. While Section 4.2 covers the impact of self-optimization steps, the average number of iterations required to achieve the optimization results in Table 1 is not clearly disclosed. This information is critical for comprehending the efficiency and practicality of the proposed method. Automating the stopping criteria for iterative optimization is also important in practice, but it is not supported by the proposed method, thus limiting its applicability.
- The potential semantic degradation in code correctness after refinement, particularly in scenarios where test cases do not cover all edge cases (which are common in real-world applications), raises concerns about the reliability of the optimized code, particularly in safety-critical scenarios.
- The methodology section, particularly section 3.4, is overly verbose. The explanation of each part of the prompt could be condensed to improve readability.

**Questions:**

## Questions & Suggestions:

1. What is the average number of iterations required for optimization? What are the stopping criteria for iterative optimization? Did you set it to 5 for all the experiments? How is it determined when to stop the refinement process?
2. How significant is the profiling overhead? Is there any analysis on the average number of extra tokens required for including the profiling information as prompts? How does this affect the overall efficiency and effectiveness of the optimization process?
3. The downgraded pass@1 results in Section 4.4 suggest that the proposed method cannot guarantee semantic equivalence after code refinement. This could impact the practicality and reliability of the code superoptimizer, especially in safety-critical applications. How can this issue be mitigated?
4. Analysis of the optimization techniques applied by the LLM is necessary. Are there certain types of optimization where the method may fall short?
5. The current approach relies on the LLM to analyze and identify hot spots from the profiling results. Given the standardized format of profiling results, could rule-based post-processing techniques that filter out hot spot information improve or accelerate the self-refinement process?
6. In line 189, only code that passed all test cases is considered in the evaluation. What if the refined code fails the test cases during optimization? How is this situation handled?
7. In Table 2, the optimization effect on memory usage appears limited compared to execution time. Also, in the ablation studies (Table 3), self-optimization with only execution time profiling feedback achieves significant memory usage reduction. Does this mean that memory optimization is relatively easier in the evaluated benchmark, or does execution time information play a more critical role in optimization?

---
[1] Madaan, Aman, Niket Tandon, Prakhar Gupta, Skyler Hallinan, Luyu Gao, Sarah Wiegreffe, Uri Alon et al. "Self-refine: Iterative refinement with self-feedback." Advances in Neural Information Processing Systems 36 (2024).

[2] Olausson, Theo X., Jeevana Priya Inala, Chenglong Wang, Jianfeng Gao, and Armando Solar-Lezama. "Is Self-Repair a Silver Bullet for Code Generation?." In The Twelfth International Conference on Learning Representations. 2023.

[3] Chang, Pohua P., Scott A. Mahlke, and Wen‐Mei W. Hwu. "Using profile information to assist classic code optimizations." Software: Practice and Experience 21, no. 12 (1991): 1301-1321.

[4] Agakov, Felix, Edwin Bonilla, John Cavazos, Björn Franke, Grigori Fursin, Michael FP O'Boyle, John Thomson, Marc Toussaint, and Christopher KI Williams. "Using machine learning to focus iterative optimization." In International Symposium on Code Generation and Optimization (CGO'06), pp. 11-pp. IEEE, 2006.

**Limitations:**

The limitations related to the overhead of profiling and the semantic correctness of the optimized code are mentioned in the paper but not analyzed in detail. A more comprehensive cost-benefit analysis, including the extra tokens required for profiling information in prompts, would provide a clearer picture of the method's overall efficiency. The authors should also consider the broader implications of this optimization approach, including its potential impact on code reliability and safety in practical applications.

---

> ### Author Rebuttal · Authors · 2024-08-06
>
> Thank you for your thoughtful review and insightful comments.
>
> **W1 & Q1: Impact of self-optimization steps**
>
> In our paper, the self-optimization steps are **constraint as 5** Section 4.2, consistent in our experiments. 5 is the default setting for many existing self-optimization methods [1, 2].
>
> [1] Islam M A, et al. MapCoder: Multi-Agent Code Generation for Competitive Problem Solving[J]. arXiv preprint arXiv:2405.11403, 2024.
>
> [2] Zhou, Andy, et al. "Language agent tree search unifies reasoning acting and planning in language models." arXiv preprint arXiv:2310.04406 (2023).
>
> Based on your suggestion, we require GPT-3.5-turbo-0301 to consider stopping the SOAP self-optimization process and only constrain the max step as 5. As shown in the Table below, the adaptive self-optimization process improves the efficiency of LLM-generated code. However, it does not achieve SOTA performance compared with SOAP. For example, when optimizing its initially generated code, it still needs 0.35s to execute, on average. SOAP only requires 0.28s. One key reason is that in each step, the efficiency of most code can be optimized. If the self-optimization stops early, the code’s efficiency remains in the sub-optimal results.
>
> | Steps | ET (s) | NET | MU (Mb) | NMU | TMU (Mb*s) | NTMU | Step |
> |-------|-----|-----|---|-----|---|--|---|
> |InitialVersion|0.36|2.50|91.25|2.45|157.50|19.75|0|
> |Adaptive|0.35|2.47|55.00|1.52|178.98|23.83|1.2|
> |SOAP|0.28|2.01|36.08|0.99|12.43|1.64|5|
>
> **W3 & Q3: Semantic degradation in code correctness after refinement where test cases do not cover all edge cases.**
>
> We clarify that most edge cases have been covered in SOAP. For EffiBench, the private test cases covered most edge cases for each canonical solution. For the other two datasets, HumanEval and MBPP, we also utilize the test cases generated by EvalPlus to cover more edge cases.
>
> On the other hand, the degradation is small (0.5% pass@1 for EffiBench, 0% pass@1 for HumanEval+, and 0.84% pass@1 for MBPP+ on average as follows). We believe that the potential degradation in correctness after refinement is manageable.
>
> ||Initial pass@1|SOAP pass@1|
> |---|----|---|
> |OpenCodeInterpreter-1.3B|30.5|30.5|
> |CodeLlama-7b|39.6|39.6|
> |CodeLlama-70b|45.7|45.7|
> |gpt-3.5-turbo|62.8|62.8|
> |claude-3-sonnet|59.1|59.1|
> *Results on HumanEval+.*
>
> | | Initial pass@1 | SOAP pass@1 |
> |---|---|----|
> |OpenCodeInterpreter-1.3B|57.9|55.8|
> |CodeLlama-7b|40.7|40.7|
> |CodeLlama-70b|53.9|52.1|
> |gpt-3.5-turbo|59.8|59.5|
> |claude-3-sonnet|58.4|58.4|
> *Results on MBPP+.*
>
> **W4: Section 3.4, is verbose**
>
> Thanks for your suggestion. We will revise it accordingly.
>
> **Q2: Analysis of the profiling overhead**
>
> We provide the breakdown of the overhead for SOAP and other prompt engineering in **Author Rebuttal Table 4**. Compared with the initial version, SOAP would increase the total token usage from 1.1M to 5.4M.
>
> **Q4: Certain types of optimization where the method may fall short**
>
> There are certain types of optimization where the SOAP method may fall short. Specifically, tasks that **do not require algorithmic improvements** or **have already been implemented with optimal time/space complexity** may not benefit significantly from SOAP. One example is that as shown in Paper Fig. 12 to Fig. 18, when LLM first generates code for the task to sort two arrays, if the code already has an optimal time/space complexity, SOAP may not optimize the code into a more efficient version. We will add this discussion in the revised version.
>
> **Q5: Could rule-based post-processing techniques improve or accelerate the self-refinement?**
>
> Based on your suggestion, we conduct experiments for the SOAP by extracting the Top 5 lines with the highest execution time and memory usage and then use these profiler parts to guide LLM to optimize its previously generated inefficient code.
>
> As shown in the Table below, rule-based optimization improves the efficiency of LLM-generated code compared with the initial version. For example, the average execution time (ET) of GPT-3.5-turbo-0301 decreases from 0.68s to 0.50s.
>
> | Steps| ET (s) | NET| MU (Mb) | NMU  | TMU (Mb*s) | NTMU  |
> |---|---|---|---|---|---|---|
> |InitialVersion|0.68|4.83|102.39|2.82|302.04|39.98|
> |Rule-Optimized|0.50|3.64|57.55|1.50|23.36|3.02|
> |SOAP|0.39|2.77|42.67|1.18|18.57|2.46|
>
> *Results for GPT-3.5-turbo-0301*
>
> However, the efficiency is lower than SOAP. For example, SOAP further decreases the average execution time for GPT-3.5-turbo-0301 from 0.50s to 0.39s. Only providing part of the overhead profile information may not make LLMs understand how to optimize the code to improve efficiency as the efficiency problems are usually based on all code.
>
> **Q6: Refined code fails the test cases during optimization?**
>
> In our experiments, we follow a two-step process:
>
> First, if the refined code does not pass the public test cases during the optimization process, we directly use the initial version of the LLM-generated code for that particular task.
>
> Second, if the code passes the public tests but fails the private tests, we filter out the entire task from our experiments for the respective model and do not calculate the optimized code's efficiency for any step of that task.
>
> For example, if step-3 optimized code is correct, but step-4 optimized code is incorrect in private test cases, we will not calculate the efficiency for any step of that task for the corresponding model, ensuring that we only consider code that passes all test cases in the evaluation and provide a fair comparison across different optimization steps and models.
>
> **Q7: Memory optimization is easier in the benchmark**
>
> One reason for memory usage being hard to optimize is that most of the tasks already achieve SOTA memory usage compared with execution time. Specifically, as seen in Table 1 in Section 4.2, most of the experiments achieve 1.00 NMU for LLM-generated initial code, which causes the code to be hard to further optimize compared with execution.

---

> > ### Comment · Reviewer_5c1j · 2024-08-12
> >
> > Thank you for the detailed response and the supplementary experiment results. I have a few follow-up questions:
> >
> > 1. Could the optimization performance be further improved with steps greater than 5? Is there a way to determine convergence in self-refinement, especially considering that the number of iterations affects token usage and LLM inference costs (as in W2)?
> > 2. In your response to Q6, you mentioned that the entire task is filtered out if the refined code fails any private test. Can I interpret the difference in your answer to W3 & Q3 (Initial pass@1 v.s. SOAP pass@1) as the proportion of filtered tasks?
> > 3. I do not think it is reasonable to filter out failed code in your evaluation. In real-world scenarios, private cases are supposed to be invisible.

---

> ### Author Response · Authors · 2024-08-13
> **Response Reviewer 5c1j follow up questions**
>
> Thank you for your response to our rebuttal.
>
> We have conducted additional experiments and provided further clarification to address the concerns raised in the follow-up questions. We hope that these efforts have adequately addressed Reviewer 5c1j's concerns. If Reviewer 5c1j thinks that the issues have been satisfactorily resolved, we would greatly appreciate it if they could consider improving the overall assessment for SOAP, as the current average overall assessment is in a borderline scenario.
>
> **Q1**
>
>
> To address your concern, we first optimized the code based on the steps outlined in Paper Table 2 and then continued to optimize for an additional five steps to analyze the efficiency of LLM-generated code after a total of 10 optimization steps. Our evaluation results are shown below. We can observe that increasing the number of optimization steps decreases the overhead, but after five steps, the efficiency metric improvements are minimal. Therefore, in our paper, we mainly set the number of steps to 5. However, if developers want to optimize their code further, they can increase the optimization steps at the cost of some additional token usage. We believe this token usage is worthwhile, as the optimized code would be deployed in the software and run thousands to millions of times.
>
> | Steps | ET (s) | NET | MU (Mb) | NMU | TMU (Mb*s) | NTMU |
> |-------|--------|-----|---------|-----|------------|------|
> | CodeLlama-70B |  |  |  |  |  |  |
> | 0     | 0.36   | 2.50 | 91.25   | 2.45 | 157.50     | 19.75 |
> | 1     | 0.33   | 2.35 | 36.09   | 0.99 | 13.70      | 1.81 |
> | 2     | 0.31   | 2.18 | 36.09   | 0.99 | 13.04      | 1.72 |
> | 3     | 0.29   | 2.06 | 36.08   | 0.99 | 12.57      | 1.66 |
> | 4     | 0.29   | 2.03 | 36.08   | 0.99 | 12.50      | 1.65 |
> | 5     | 0.28   | 2.01 | 36.08   | 0.99 | 12.43      | 1.64 |
> | 6     | 0.27   | 2.00 | 36.07   | 0.99 | 12.40      | 1.63 |
> | 7     | 0.27   | 1.98 | 36.06   | 0.99 | 12.37      | 1.62 |
> | 8     | 0.27   | 1.98 | 36.05   | 0.99 | 12.35      | 1.61 |
> | 9     | 0.27   | 1.97 | 36.04   | 0.99 | 12.35      | 1.60 |
> | 10    | 0.26   | 1.96 | 36.03   | 0.99 | 12.35      | 1.59 |
> | GPT-3.5-Turbo-0301 |  |  |  |  |  |  |
> | 0     | 0.36   | 2.50 | 91.25   | 2.45 | 157.50     | 19.75 |
> | 1     | 0.33   | 2.35 | 36.09   | 0.99 | 13.70      | 1.81 |
> | 2     | 0.31   | 2.18 | 36.09   | 0.99 | 13.04      | 1.72 |
> | 3     | 0.29   | 2.06 | 36.08   | 0.99 | 12.57      | 1.66 |
> | 4     | 0.29   | 2.03 | 36.08   | 0.99 | 12.50      | 1.65 |
> | 5     | 0.28   | 2.01 | 36.08   | 0.99 | 12.43      | 1.64 |
> | 6     | 0.28   | 2.00 | 36.07   | 0.99 | 12.40      | 1.63 |
> | 7     | 0.27   | 1.99 | 36.06   | 0.99 | 12.38      | 1.62 |
> | 8     | 0.27   | 1.98 | 36.05   | 0.99 | 12.36      | 1.61 |
> | 9     | 0.27   | 1.97 | 36.04   | 0.99 | 12.35      | 1.60 |
> | 10    | 0.27   | 1.97 | 36.04   | 0.99 | 12.33      | 1.59 |
>
> **Q2**
>
> Response:
>
> **The difference between Initial pass@1 v.s. SOAP pass@1 is only the SOAP-generated incorrect code.** While the address method discussed in Q6 does not reflect in W3 & Q3, which means that with our solution in Q6, the incorrect code generated by SOAP in SOAP pass@1 will be replaced by the Initial generated code.
>
>
> **Q3.**
>
> We understand your concern about filtering out failed code in our evaluation. However, our approach is consistent with existing works in this field [1-6]. The primary reason for focusing only on correct code is that incorrect code may cause test cases to fail and terminate the execution process prematurely. Including such cases would lead to inaccurate results when measuring the efficiency of LLM-generated code.
>
> [1] Huang, Dong, Jie M. Zhang, Yuhao Qing, and Heming Cui. "EffiBench: Benchmarking the Efficiency of Automatically Generated Code." arXiv preprint arXiv:2402.02037 (2024).
>
> [2] Qiu, Ruizhong, Weiliang Will Zeng, Hanghang Tong, James Ezick, and Christopher Lott. "How Efficient is LLM-Generated Code? A Rigorous & High-Standard Benchmark." arXiv preprint arXiv:2406.06647 (2024).
>
> [3] Shi, Jieke, Zhou Yang, and David Lo. "Efficient and Green Large Language Models for Software Engineering: Vision and the Road Ahead." arXiv preprint arXiv:2404.04566 (2024).
>
> [4] Zheng, Jiasheng, Boxi Cao, Zhengzhao Ma, Ruotong Pan, Hongyu Lin, Yaojie Lu, Xianpei Han, and Le Sun. "Beyond Correctness: Benchmarking Multi-dimensional Code Generation for Large Language Models." arXiv preprint arXiv:2407.11470 (2024).
>
> [5] Waghjale, Siddhant, Vishruth Veerendranath, Zora Zhiruo Wang, and Daniel Fried. "ECCO: Can We Improve Model-Generated Code Efficiency Without Sacrificing Functional Correctness?." arXiv preprint arXiv:2407.14044 (2024).
>
> [6] Du, Mingzhe, Anh Tuan Luu, Bin Ji, and See-Kiong Ng. "Mercury: An efficiency benchmark for llm code synthesis." arXiv preprint arXiv:2402.07844 (2024).

---

> ### Author Response · Authors · 2024-08-14
> **Friendly Reminder: Pending Reviewer Responses and Assessment Consideration**
>
> Dear Reviewer 5c1j,
>
> Thank you for your review and comments on our paper on 13 Aug, we have provided the clarification and other experiment results to address your concern.
>
> Since the discussion period will be closed in the next six hours, while we do not receive the message about whether we have addressed all of your concerns, we would greatly appreciate it if you could take the time to review these results and consider improving your overall assessment of our paper based on these revisions.
>
> Thank you once again for your valuable feedback and for considering our request.

---

### Official Review · Reviewer_H3VZ · 2024-07-11

**Soundness:** 2
**Presentation:** 3
**Contribution:** 2
**Rating:** 3
**Confidence:** 5

**Summary:**

This paper presents SOAP, a new code generation approach that improves the efficiency of code generated by a LLM. SOAP adopts a self-refinement method that iteratively prompts the LLM to re-generate the code based on the profiling results. Specifically, it uses the line-profiler package in Pyhton to get the execution time profiling results and the memory_profiler package to get the memory usage profiling result. SOAP is evaluated on three benchmarks and 22 LLMs, including both 16 open-sourced and 6 closed-sourced models. The results show that SOAP can effectively improve the execution time and memory usage of code generated by most LLMs. While it slightly decreases the LLM's performance on functional correctness, the impact is small.

**Strengths:**

1. SOAP is evaluated on multiple benchmarks and many LLMs.
2. The results look promising.
3. The paper is well-written and easy to follow.

**Weaknesses:**

1. The technical novelty of this work is limited. The self-refinement method proposed in this work is very similar to existing self-refinement methods such as Self-Edit and Critic. The only difference is that SOAP uses profiling results while existing methods use testing results, etc.
2. The authors ignored existing work on using transformers or LLMs for code optimization. They should discuss these papers in the related work and also use them as comparison baselines.
- Liu, Shengcai, et al. "Large language models as evolutionary optimizers." arXiv preprint arXiv:2310.19046 (2023).
- Chen, Zimin, Sen Fang, and Martin Monperrus. "Supersonic: Learning to generate source code optimisations in c/c++." arXiv preprint arXiv:2309.14846 (2023).
- Shypula, Alexander, et al. "Learning performance-improving code edits." arXiv preprint arXiv:2302.07867 (2023).
3. Section 4.3 compares SOAP with Self-Refine and Reflexion. It is unclear how Self-Refine and Reflexion were configured in this experiment. Did the authors use the original prompts from Self-Refine and Reflexion? Or did the authors modify their prompts to include instructions on code efficiency? If the former, there is a comparison fairness issue since the original prompts in Self-Refine and Reflexion are not designed for code optimization. If the latter, the authors should provide the prompts used in Self-Refine and Reflexion and justify the prompt design.
4. The experiments on the impact of step-optimization steps (Section 4.2) only include two big models. The findings may not be generalizable to smaller models. Why not also include some smaller models?
5. The experiments on the impact of functional correctness were only done on EffiBench. Since EffiBench is not specifically designed for
 functional code generation, the results may not be representative. The authors should also experiment with HumanEval and MBPP and report the pass@1 scores on these two benchmarks.

**Questions:**

1. What is the key technical novelty of this work?
2. How would you compare SOAP with existing transformer or LLM-based code optimization work?
3. What are the prompts used in the experiments with Self-Refine and Reflexion?

**Limitations:**

The discussion on limitations looks reasonable.

---

> ### Author Rebuttal · Authors · 2024-08-06
>
> Thank you for your thoughtful review and insightful comments. We hereby address your concerns below:
>
> **W1 & Q1: Novelty and Similarities with Self-Edit and Critic**
>
> To address the reviewer’s concern about the novelty of SOAP in comparison to Self-Edit and Critic, we provide detailed evaluation results in Table below. These results demonstrate that SOAP outperforms both Self-Edit and Critic significantly.
>
> | Optimization Profile | ET (s) | NET | MU (Mb) | NMU | TMU (Mb*s) | NTMU |
> |---|---|---|---|---|---|---|
> |**GPT-3.5-Turbo-0301**|||||||
> |InitialVersion|0.36|2.50|91.25|2.45|157.50|19.75|
> |Self-Edit|0.42|3.67|59.86|1.65|24.87|3.28|
> |Critic|0.60|4.25|102.75|2.82|351.91|46.39|
> |SOAP|0.28|2.01|36.08|0.99|12.43|1.64|
>
> SOAP introduces a unique overhead profiler that significantly enhances the efficiency of LLM-generated code by optimizing both execution time and memory usage. In contrast, existing methods focus primarily on execution time and cannot comprehensively address efficiency concerns. Specifically, Critic requires progressive amendment of outputs, which does not effectively tackle efficiency-critical issues.
>
> The novelty of SOAP lies in two aspects:
>
> - **Dual Optimization Focus**: Unlike existing works that primarily target execution time, SOAP optimizes both execution time and memory usage, providing a more holistic approach to code efficiency.
>
> - **Overhead Profile-Guided Optimization**: SOAP utilizes overhead profiles, including execution time and memory usage profiles, to guide the optimization process. This allows SOAP to achieve SOTA results in efficiency, as demonstrated by our ablation studies.
>
> We would like to highlight that self-refinement is a broad direction with many published papers exploring various novelties. The specific challenge addressed by SOAP is improving the efficiency of LLM-generated code, a critical and valuable contribution to the field. Our experiments clearly show that the inclusion of overhead profiles significantly enhances optimization effectiveness compared to existing methods.
>
> **W2 & Q2: Discuss and compare transformers or LLMs for code optimization.**
>
> We would like to highlight that SOAP and these methods [1,2,3] are orthogonal, which means we can combine SOAP and these efforts.
>
> LMEA [1] uses evolutionary search to generate more efficient code compared with the initial code, which requires 250+ solutions for each task to achieve optimal results, whereas SOAP only generates 5 solutions. LMEA requires massive token usage and execution time consumption. Due to limited time and resources during rebuttal,  we are not able to conduct experiments with LMEA. However, we commit to include the results for LMEA in the revised manuscript.
>
> For Supersonic [2] and PIE [3], we conduct experiments in **Author Rebuttal Table 1**. For PIE, we conduct experiments for five different prompts. As shown in Table 1, the performance of these methods is further boosted when SOAP is applied to them. On average, PIE+Few-Shot requires 0.82s for each generated code to execute. While PIE+SOAP+Few-Shot generated code only requires 0.41s to execute.
>
> Ref:
>
> [1] Large language models as evolutionary optimizers. 2023
>
> [2] Supersonic: Learning to generate source code optimizations in c/c++. 2023
>
> [3] Learning performance-improving code edits. 2023
>
>
> **W3 & Q3: Prompts for Self-Refine and Reflexion**
>
> For a fair comparison, we conduct experiments for baselines by revising the prompt used by SOAP in our experiments, which specifically requires LLMs to refine the coding efficiency. We provide the detailed prompts used in the paper in the Link: https://anonymous.4open.science/r/SOAP-FF6C/prompts.py
>
>
> **W4: Step-optimization steps only include two big models.**
>
> We provide three smaller models, i.e., OpenCodeInterpreter-1.3B, DeepSeek-1.3B-Ins, and CodeLlama-7B below. We can observe that SOAP consistently optimizes the efficiency of LLM-generated code for each optimization step.
>
> | Steps | ET (s) | NET | MU (Mb) | NMU | TMU (Mb*s) | NTMU |
> | --- | --- | --- | --- | --- | --- | --- |
> |**OpenCodeInterpreter-1.3B**|||||||
> |0|1.60|1.52|38.91|1.00|89.16|1.11|
> |1|1.60|1.51|38.91|1.00|88.77|1.11|
> |2|1.59|1.51|38.91|1.00|88.16|1.10|
> |3|1.59|1.50|38.91|1.00|87.85|1.09|
> |4|1.59|1.50|38.91|1.00|87.77|1.09|
> |5|1.29|1.23|38.91|1.00|70.63|0.88|
> |**DeepSeek-1.3B-Ins**|||||||
> |0|1.42|1.32|36.04|1.00|40.61|1.12|
> |1|1.42|1.23|36.04|1.00|39.30|1.09|
> |2|1.42|1.20|36.04|1.00|37.99|1.05|
> |3|1.15|1.16|36.04|1.00|37.33|1.03|
> |4|1.15|1.12|36.04|1.00|36.43|1.00|
> |5|1.15|1.07|36.04|1.00|35.48|0.98|
> |**CodeLlama-7B**|||||||
> |0|4.70|3.68|46.76|0.99|212.41|1.93|
> |1|4.59|3.58|38.94|0.82|165.16|1.50|
> |2|4.56|3.55|39.11|0.82|162.16|1.47|
> |3|4.56|3.56|39.03|0.82|161.98|1.47|
> |4|4.56|3.55|39.03|0.82|161.57|1.47|
> |5|4.52|3.54|38.67|0.82|157.76|1.43|
> *SOAP for different optimization steps.*
>
> **W5: Pass@1 on HumanEval and MBPP.**
>
> As shown in the tables below, we provide the pass@1 for HumanEval+ and MBPP+, where the evalplus version is used in our experiments to measure the efficiency of LLM-generated code as it contains hundreds of tests. We can observe that the pass@1 of LLM-generated code **only has merely changed for these benchmarks**. The performance of our evaluated LLMs on HumanEval+ does not show a decrease. On MBPP+, the pass@1 decrease is only 0.0% and 0.84% for HumanEval+ and MBPP+, on average. Notably, for more powerful LLMs like GPT and Claude, the performance remains essentially unchanged.
>
> | Model | Initial pass@1 | SOAP pass@1 |
> |---|----|----|
> |OpenCodeInterpreter-DS-1.3B|30.5|30.5|
> |CodeLlama-7b-Instruct-hf|39.6|39.6|
> |CodeLlama-70b-Instruct-hf|45.7|45.7|
> |gpt-3.5-turbo-030|62.8|62.8|
> |claude-3-sonnet|59.1|59.1|
> *Results on HumanEval+.*
>
> | Model | Initial pass@1 | SOAP pass@1 |
> |-------|-----|------|
> |OpenCodeInterpreter-DS-1.3B|57.9|55.8|
> |CodeLlama-7b-Instruct-hf|40.7|40.7|
> |CodeLlama-70b-Instruct-hf|53.9|52.1|
> |gpt-3.5-turbo-0301|59.8|59.5|
> |claude-3-sonnet|58.4|58.4|
> *Results on MBPP+.*

---

> ### Author Response · Authors · 2024-08-14
> **Friendly Reminder: Pending Reviewer Responses and Assessment Consideration**
>
> Dear Reviewer H3VZ,
>
> Thank you for your review and comments on our paper.
>
> Since the discussion period will be closed in the next six hours, while we do not receive the message about whether we have addressed all of your concerns, we would greatly appreciate it if you could take the time to review these results and consider improving your overall assessment of our paper based on these revisions.
>
> Thank you once again for your valuable feedback and for considering our request.

---

> > ### Comment · Reviewer_H3VZ · 2024-08-14
> > **Response to the rebuttal**
> >
> > Thank you for the responses, and sorry for the late response due to a flurry of proposal and review dues. Regarding the novelty, would it be fair to say the essential difference to self-refine is the inclusion of the profiling results from the `line_profiler` library in the prompt?
> >
> > While I appreciate the authors' effort in conducting new experiments quickly in the short rebuttal period, I feel these results require another round of careful reviews, especially given that some results look quite surprising and counterintuitive. Without knowing more details and experiment settings, it is hard to make a full assessment of the correctness of the results.
> >
> > In particular, it's surprising to see that the pass@1 rates of all these models barely change after applying SOAP. Note that compared with the original code generation prompt, SOAP adds a lot of line-level profiling results, which significantly deviates from the original prompt. Presumably, we should see some deviations in pass@1, as LLMs are pretty sensitive to prompt design and including many profiling results may distract the LLM from reading the original task description. How did the authors keep the pass@1 rates almost unchanged?
> >
> > Furthermore, the evaluation results of PIE with CodeLlama7B look quite strange. The average execution time of code optimized by PIE is in the range of 0.4s to 0.9s. However, the execution time of CodeLlama7B in Table 1 is 4.7s before SOAP optimization and 4.52s after SOAP optimization. If the execution time after PIE optimization is correct, **does this mean PIE can achieve 10x speed up compared with the initial code and the code optimized by SOAP alone?** If that's the case, what's the point of applying SOAP on top of PIE given that PIE has already achieved such a big improvement?
> >
> > Overall, I feel this paper requires another round of careful reviews before it can be accepted.

---

> ### Author Response · Authors · 2024-08-14
> **Response to Reviewer H3VZ**
>
> Dear Reviewer H3VZ,
>
> Thanks for your response before the discussion period deadline, which allows us to further clarify the potential misunderstanding and concerns that would affect your overall assessment.
>
> First, compared with self-refine, SOAP utilizes line_profiler and memory profiler to catch the execution time and memory usage of each line in the LLM-generated code.
>
> Second, for the pass@1 of LLM-generated code of pre-SOAP and post-SOAP, we want to mention that in https://anonymous.4open.science/r/SOAP-FF6C/src/SOAP.py (Lines 177-Lines 185), we provide tolerance for SOAP that requires the SOAP optimized code can pass public test cases and then replace original code. If the optimized code can not pass the public test cases, SOAP optimized code will not replace the original code, and we will use the original code for the next step of optimization (Lines 184-185). In this step, the coverage of the public test cases would largely affect the correctness of the final SOAP-optimized code. After the empirical study, we observe that the code line coverage of EffiBench for public test cases achieves **99.24%**, which makes sure that the pass@1 after the self-optimization process (SOAP) only has a few decreases, and even in some models, the pass@1 would not decrease.
>
> Third, we need to mention that PIE already trained CodeLlama with its provided code, which means that the pass@1 of PIE+CodeLlama-7B may be different, and the tasks addressed by PIE and the original CodeLlama are different. Then, some tasks with higher execution time (e.g., 4.7s) may not addressed by PIE-CodeLlama, and the default (initial) ET and other metrics are different. We hope that Reviewer H3VZ can understand this difference between the original CodeLlama and PIE-CodeLlama.
>
> We hope that our clarification can address Reviewer H3VZ's concern about **I feel these results require another round of careful reviews, especially given that some results look quite surprising and counterintuitive.**

---

### Author Rebuttal · Authors · 2024-08-06

# Tables included in the rebuttal to address the comments made by Reviewer H3VZ, Reviewer VC11, and Reviewer pc3i

|OptimizationProfile|ET(s)|NET|MU(Mb)|NMU|TMU(Mb*s)|NTMU|
|---|---|---|---|---|---|---|
|**GPT-3.5-Turbo-0301**|||||||
|InitialVersion|0.36|2.50|91.25|2.45|157.50|19.75|
|UnsupervisedSelf-Refine|0.32|2.46|78.39|2.12|312.99|42.42|
|Result-AwareSelf-Refine|0.30|2.25|58.65|1.61|195.49|27.16|
|Self-Edit|0.42|3.67|59.86|1.65|24.87|3.28|
|Critic|0.60|4.25|102.75|2.82|351.91|46.39|
|DirectlyEfficiency|0.43|3.03|59.11|1.67|20.37|2.88|
|Self-RefineEfficiency|0.40|2.83|59.11|1.67|18.80|2.65|
|IsSelf-Refine|0.40|2.88|61.83|1.81|36.29|5.69|
|Self-Reasoning|0.89|6.21|60.64|1.62|45.91|5.61|
|Self-Relfection|0.81|5.67|60.64|1.62|39.35|4.80|
|SOAP|0.28|2.01|36.08|0.99|12.43|1.64|
|**CodeLlama7B(PIE:HQ+SelfPlay)**|||||||
|PIE+Zero-Shot|0.87|5.73|74.83|1.81|109.29|9.69|
|PIE+SOAP+Zero-Shot|0.79|5.41|65.78|1.68|89.90|7.84|
|PIE+Few-Shot|0.82|5.58|73.57|1.74|98.02|8.92|
|PIE+SOAP+Few-Shot|0.41|2.97|73.10|1.74|59.69|5.09|
|PIE+CoT|0.79|5.14|73.14|1.74|63.93|5.35|
|PIE+SOAP+COT|0.45|2.84|71.15|1.71|58.06|4.77|
|PIE+DynamicRetrieval,k=4|0.74|5.36|68.64|1.51|85.24|7.78|
|PIE+SOAP+DynamicRetrieval,k=4|0.41|3.36|68.63|1.51|52.34|4.52|
|**Supersonic**|||||||
|Supersonic|1.40|10.33|113.06|3.18|329.59|56.24|
|Supersonic+SOAP|1.34|9.91|102.26|2.87|267.47|45.64|

*Author Rebuttal Table 1: Evaluation results of SOAP and baselines. Since the finetuned link for GPT-3.5-turbo from PIE is not available, we use the finetuned CodeLlama 7B for experiments.*

|Steps|ET(s)|NET|MU(Mb)|NMU|TMU(Mb*s)|NTMU|
|-------|--------|-----|---------|-----|------------|------|
|**gpt-3.5-turbo-0301**|||||||
|InitialVersion|0.37|3.10|66.91|1.90|20.89|6.32|
|SOAP|0.28|2.31|67.31|1.92|15.66|4.71|
|**gpt-3.5-turbo-0613**|||||||
|InitialVersion|0.37|3.05|66.99|1.90|20.92|6.18|
|SOAP|0.32|2.69|67.09|1.91|18.40|5.45|
|**gpt-3.5-turbo-1106**|||||||
|InitialVersion|0.37|3.07|66.94|1.90|20.78|6.22|
|SOAP|0.32|2.72|66.98|1.91|18.29|5.54|
|**gpt-4**|||||||
|InitialVersion|0.37|3.06|66.91|1.90|21.17|6.17|
|SOAP|0.32|2.69|66.97|1.91|18.42|5.36|
|**gpt-4-turbo-preview**|||||||
|InitialVersion|0.37|3.10|66.92|1.90|20.78|6.28|
|SOAP|0.32|2.67|67.01|1.91|18.65|5.42|
|**claude-3-haiku**|||||||
|InitialVersion|0.39|3.27|66.90|1.90|22.52|6.68|
|SOAP|0.32|2.65|67.02|1.91|18.83|5.42|
|**claude-3-sonnet**|||||||
|InitialVersion|0.38|3.20|66.93|1.90|21.52|6.55|
|SOAP|0.32|2.65|66.95|1.91|18.30|5.43|

*Author Rebuttal Table 2: Evaluation results for five different LLMs on identical tasks verified by correct execution (210 tasks).*

|Steps|ET(s)|NET|MU(Mb)|NMU|TMU(Mb*s)|NTMU|
|-------|--------|-----|---------|-----|------------|------|
|**OpenCodeInterpreter-1.3B**|||||||
|InitialVersion|0.80|5.74|62.60|1.61|45.18|5.63|
|SOAP|0.52|3.73|48.73|1.25|29.93|3.73|
|**deepseek-coder-1.3b-instruct**|||||||
|InitialVersion|0.63|4.57|59.32|1.64|27.57|5.49|
|SOAP|0.43|3.10|45.80|1.27|17.44|3.47|
|**CodeLlama-7b-Instruct-hf**|||||||
|InitialVersion|3.98|23.93|103.78|2.19|370.74|28.06|
|SOAP|3.41|20.50|89.37|1.88|335.53|25.39|
|**CodeLlama-70b-Instruct-hf**|||||||
|InitialVersion|0.90|7.34|50.69|1.91|46.93|21.07|
|SOAP|0.63|5.18|41.44|1.56|34.90|15.67|
|**starcoder2-15b**|||||||
|InitialVersion|0.49|1.88|132.27|1.17|53.54|1.18|
|SOAP|0.22|0.86|66.18|0.59|25.24|0.56|
|**gpt-3.5-turbo-0301**|||||||
|InitialVersion|0.68|4.83|102.39|2.82|302.04|39.98|
|SOAP|0.39|2.77|42.67|1.18|18.57|2.46|
|**claude-3-sonnet**|||||||
|InitialVersion|0.74|5.20|64.02|1.76|56.85|7.29|
|SOAP|0.51|3.60|46.88|1.29|27.24|3.49|

*Author Rebuttal Table 3: Evaluation results of SOAP and baselines in PIE provided default simulator. We use the default simulator provided by PIE for our experiments.*

|Model|ExecutionTime(s)|TokenUsage(million)|PerIterationInput/OutputTaskTokenUsage(K)|
|-------|-------------------|----------------------|--------------------------------------------------|
|InitialVersion|76.81|1.1|1.1|
|UnsupervisedSelf-Refine|416.81|3.9|1.1602|
|Result-AwareSelf-Refine|419.24|3.9|1.1602|
|LineProfiler|555.68|4.7|1.4888|
|MemoryProfiler|536.26|4.7|1.4888|
|SOAP|566.67|5.4|1.7805|

*Author Rebuttal Table 4: Overhead of different code efficiency optimization prompt engineering methods for GPT-3.5-turbo.*

|Steps|CoV of ET(%)|CoV of NET(%)|CoV of MU(%)|CoV of NMU(%)|CoV of TMU(%)|CoV of NTMU(%)|
|-------|---------------|----------------|---------------|----------------|----------------|-----------------|
|OpenCodeInterpreter-DS-1.3B|2.5|2.5|2.5|2.5|2.5|2.5|
|deepseek-coder-1.3b-instruct|1.7|1.7|1.7|1.7|1.7|1.7|
|CodeLlama-7b-Instruct-hf|0.3|0.3|0.3|0.3|0.3|0.3|
|CodeLlama-70b-Instruct-hf|1.7|1.7|1.7|1.7|1.7|1.7|
|starcoder2-15b|1.6|1.6|1.6|1.6|1.6|1.6|
|gpt-3.5-turbo-0301|0.6|0.6|0.6|0.6|0.6|0.6|
|claude-3-sonnet|0.2|0.2|0.2|0.2|0.2|0.2|

*Author Rebuttal Table 5: Overall coefficient of variation from five executions of SOAP for each metric and model in EffiBench. We conducted the experiments five times, then calculated the mean and std for each metric and model.*

| Steps | ET (s) | NET | MU (Mb) | NMU | TMU (Mb*s) | NTMU |
| --- | --- | --- | --- | --- | --- | --- |
|**OpenCodeInterpreter-1.3B**|||||||
|0|1.60|1.52|38.91|1.00|89.16|1.11|
|1|1.60|1.51|38.91|1.00|88.77|1.11|
|2|1.59|1.51|38.91|1.00|88.16|1.10|
|3|1.59|1.50|38.91|1.00|87.85|1.09|
|4|1.59|1.50|38.91|1.00|87.77|1.09|
|5|1.29|1.23|38.91|1.00|70.63|0.88|
|**DeepSeek-1.3B-Ins**|||||||
|0|1.42|1.32|36.04|1.00|40.61|1.12|
|1|1.42|1.23|36.04|1.00|39.30|1.09|
|2|1.42|1.20|36.04|1.00|37.99|1.05|
|3|1.15|1.16|36.04|1.00|37.33|1.03|
|4|1.15|1.12|36.04|1.00|36.43|1.00|
|5|1.15|1.07|36.04|1.00|35.48|0.98|
|**CodeLlama-7B**|||||||
|0|4.70|3.68|46.76|0.99|212.41|1.93|
|1|4.59|3.58|38.94|0.82|165.16|1.50|
|2|4.56|3.55|39.11|0.82|162.16|1.47|
|3|4.56|3.56|39.03|0.82|161.98|1.47|
|4|4.56|3.55|39.03|0.82|161.57|1.47|
|5|4.52|3.54|38.67|0.82|157.76|1.43|

*Author Rebuttal Table 6: Evaluation results of SOAP for different optimization step.*

---

### Comment · Area_Chair_gHRy · 2024-08-09
**Author-Reviewer discussions (Aug 7 - Aug 13)**

Dear Submission5230 reviewers,

We appreciate your reviews as we move into the Author-Reviewer discussions phase (Aug 7 - Aug 13).
Please read all reviews and author responses carefully.
Please address any remaining questions or concerns with the authors and respond to their rebuttal as needed. Authors need time to respond to your messages, so please post your responses as soon as possible, so there is time for back-and-forth discussion with the authors. At a minimum, please acknowledge that you have read the author rebuttal. Based on the material provided, please adjust your scores and reviews as necessary.

Dear Submission5230 authors,

We appreciate your rebuttal. Please continue the dialogue with the reviewers during this discussion phase.

This message and thread are visible to both reviewers and authors. If you need to respond privately, adjust the "Readers" setting accordingly.

---

### Comment · Area_Chair_gHRy · 2024-08-09
**Question for authors**

I appreciate your submission, detailed reviewer reviews, and your extensive rebuttal.

One question that I did not see discussed yet that might be important for readers and reviewers:
In Table 1, large percentage improvement is mostly achieved for very inefficient code especially for memory metrics. (StarCoder2-15B  ET is impressive exception).

You have a number of code solutions of various efficiency. It would be interesting to see the performance of SOAP for the code (not necessarily generated by the same LLM) that ranges from bad to "optimal" (the best available) solution.

Since the above would require a lot of experiments (N code variants x M LLMs), an alternative would be to provide ratio of best code metric after SOAP (irrespective of LLM) to best code before SOAP (irrespective of LLM). E.g. for ET in Table 1, this would be 0.12 / 0.29. This is cruder metric, since there is no guarantee that 0.12 can be achieved by running SOAP on 0.29 code, but it gives some measure of how much SOAP can improve good code. For example, for MU this metric shows very low improvement of 24.28 / 24.31.

---

> ### Author Response · Authors · 2024-08-10
> **Response about global optimal**
>
> Dear AC,
>
> Thanks for your efforts to remind reviewers to participate in the discussion process.
>
> To address the concern about the global efficiency results, we conduct experiments with our previously generated source code and SOAP-optimized code. We follow the below instructions to calculate the overall efficiency. First, we collect all correct LLM-generated code for each task and then select the most efficient code generated by LLMs based on ET, MU, and TMU. Then, we calculate the overall efficiency metrics for each metric-guided selected code for the initial code and SOAP-optimized code. The evaluation results for EffiBench are shown below. We can observe that in this scenario, SOAP largely improved the efficiency of LLM-generated code. For example, in the Optimizal ET results (i.e., we collect source code for both initial and SOAP optimized code for each task), SOAP decreases the average ET from 0.97 (s) to 0.23 (s). In Optimizal MU, SOAP decreases the MU from 59.16 (Mb) to 21.13 (Mb). In Optimial TMU, SOAP also decreases the TMU from 137.18 (Mb*s) to 10.14 (Mb*s). We hope our experiments can address your concern.
>
> | Steps   | ET (s) | NET   | MU (Mb) | NMU  | TMU (Mb*s) | NTMU |
> |---------|--------|-------|---------|------|------------|------|
> | Optimal ET |        |       |         |      |            |      |
> | Initial | 0.97   | 7.55  | 59.26   | 1.75 | 153.39     | 27.01|
> | SOAP | 0.23   | 1.59  | 29.74   | 1.03 | 13.71      | 1.59 |
> | Optimal MU |        |       |         |      |            |      |
> | Initial | 1.33   | 10.43 | 59.16   | 1.75 | 169.51     | 29.85|
> | SOAP | 0.74   | 4.91  | 21.13   | 0.83 | 21.22      | 1.93 |
> | Optimal TMU |        |       |         |      |            |      |
> | Initial | 1.38   | 10.95  | 59.25   | 1.75 | 137.18     | 26.13|
> | SOAP | 0.47   | 3.41  | 31.17   | 1.16 | 10.14      | 1.37 |

---

> ### Comment · Area_Chair_gHRy · 2024-08-10
> **Thank you for your response.**
>
> I appreciate your response.
> I think this is a good metric.
>
> However, could you clarify the numbers in your table?
>
> I would think that "Optimal ET" would have initial 0.29 and after-SOAP 0.25 (DeepSeek-33B-Ins and OpenCodeInterpreter-33B are most efficient based on ET pre-SOAP, I picked DeepSeek-33B-Ins since SOAP improves its code more), "Optimal MU" would have initial 24.31 and after-SOAP 24.28 (StarCoder2-3B is most efficient based on MU), "Optimal TMU" would have initial 13.06 and after-SOAP 11.54 (OpenCodeInterpreter-33B is most efficient based on TMU). Am I misunderstanding your metric? (I did not look up other columns in the table just the ones corresponding to the Optimal metric in your subheading)
>
> Thank you again.

---

> > ### Author Response · Authors · 2024-08-10
> > **Response for AC question**
> >
> > Dear Area Chair gHRy,
> >
> > Clarification for Previously Provided Results:
> >
> > Thank you for your response. We would like to address your concerns and clarify our setup as follows:
> >
> > In the experiment for optimal ET, MU, and TMU, we first gather solutions from all LLMs for each task in our EffiBench. Subsequently, we calculate the ET, MU, and TMU for each solution from the collected solutions and rank them based on these three metrics. For instance, considering ET, we select the most efficient solution (that is, a solution with the lowest ET among all model-generated codes) as the pre-SOAP solution. This represents the most effective solution in terms of ET in the pre-SOAP stage. Similarly, for SOAP, we follow the same strategy to obtain the most efficient ET solution.
> >
> > Through this process, we do not directly select the 0.29 (s) OpenCodeInterpreter-33B's pre-SOAP as the optimal ET since some tasks might not be addressed by OpenCodeInterpreter-33B. Other LLMs that may address these tasks are also used to calculate pre-SOAP. Lastly, we compute the efficiency for both the pre-SOAP most efficient code and the post-SOAP most efficient code, but only for correct codes in the experiments.
> >
> > Indeed, the method you proposed is excellent. Based on your suggestion, we have conducted the following experiments. As can be seen, in this setup, SOAP also reduces the overhead of the initial version across all metrics. For instance, the ET decreases by 58.6%, reducing the average execution time from 0.29 seconds to 0.12 seconds. In our final version, we will also provide the evaluation results of SOAP for the N code variants x M LLMs.
> >
> >
> > | Steps   | **ET (s)** | **NET** | **MU (Mb)** | **NMU** | **TMU (Mb*s)** | **NTMU** |
> > |---------|------------|---------|-------------|---------|----------------|----------|
> > | Initial | 0.29       | 1.25    | 24.31       | 0.93    | 11.84          | 1.11     |
> > | SOAP    | 0.12 (58.6\%)| 1.03 (17.6\%)| 24.28 (0.1\%)| 0.82 (11.8\%)| 2.03 (82.9\%)| 0.88 (20.7\%)|

---

> > > ### Comment · Area_Chair_gHRy · 2024-08-10
> > > **Thank you for clarification and additional results**
> > >
> > > Thank you for clarification and additional results.

---

### Decision · Program_Chairs · 2024-09-25

**Decision:**

Accept (poster)

**Comment:**

This paper has a very wide spread of scores from 3 (reject) to 8 (strong accept).
After reading the paper, reviews, rebuttal, and substantial discussions between authors and reviewers, I believe that the paper should be accepted.

There has been extended rebuttals by authors with numerous additional results as well as additional comments and questions by reviewers. I believe the key issues are:

1. Is the contribution novel enough? Adding profiling results to LLM prompt to improve the code efficiency is novel. Reviewers are split whether the novelty is significant. I would rate novelty by itself in the weak accept area.
2. Is the evaluation done well? I believe that authors have improved the evaluation significantly during the rebuttal process. I agree with reviewer VC11 that "the current evaluation of SOAP adequately demonstrates the effectiveness of the work". However, I acknowledge the reviewers who pointed out somewhat questionable results in rebuttal. I am also somewhat concerned on how the authors are going to integrate all the additional results from rebuttal into the paper and whether the evaluation will be presented well. I would rate the evaluation also at the level of weak accept (I might change it to higher if the evaluation was well integrated into the paper.)
3. Functional Correctness: Some reviewers expressed concerns about the impact on functional correctness arguing that the SOAP method might introduce semantic degradation in code correctness after refinement. Authors have shown minimal correctness degradation in the benchmarks suggested by the concerned reviewers. In my opinion the correctness degradation may occur, but it is not a blocker for adopting SOAP like techniques.

Based on my evaluation of these issues - taking into account all the reviews, rebuttal, and discussions - I propose we accept the paper.